# AUTOGETS: AUTOMATED GENERATION OF TEXT SYNTHETICS FOR IMPROVING TEXT CLASSIFICATION

## ABSTRACT

When developing text classification models for real world applications, one major challenge is the difficulty to collect sufficient data for all text classes. In this work, we address this challenge by utilizing large language models (LLMs) to generate synthetic data and using such data to improve the performance of the models without waiting for more real data to be collected and labelled. As an LLM generates different synthetic data in response to different input examples, we formulate an automated workflow, which searches for input examples that lead to more "effective" synthetic data for improving the model concerned. We study three search strategies with an extensive set of experiments, and use experiment results to inform an ensemble algorithm that selects a search strategy according to the characteristics of a class. Our further experiments demonstrate that this ensemble approach is more effective than each individual strategy in our automated workflow for improving classification models using LLMs.

## 1 INTRODUCTION

A critical impediment to developing robust text classification models for real-world applications is the pervasive challenge of class imbalance and data scarcity, particularly for underrepresented text categories. Many industrial applications, such as ticketing systems, require classification models to process large volumes of unstructured text data, such as problem descriptions and user comments, which are often heavily imbalanced in class sizes. In modern industrial environments, ticketing systems play a vital role in managing and resolving technical issues, service requests, and operational incidents (Al-Hawari & Barham (2021)). As shown in the workflow (Figure 1), models are initially trained on a set of labeled tickets, but newly introduced or infrequent classes often arise after deployment, necessitating manual classification and correction. This reliance on manual intervention for new or underrepresented classes creates operational bottlenecks and impairs model adaptation to evolving data distributions. Over time, the model's performance degrades, particularly for small or specialized categories, as obtaining balanced and adequately labeled data across all classes remains challenging. Consequently, models often fail to generalize effectively across diverse data distributions, especially for underrepresented categories Gandla et al. (2024). Traditionally, addressing data scarcity involves collecting additional real-world data, which can be both time-consuming and resource-intensive, especially for rare or newly introduced classes. Furthermore, the manual labeling of such data introduces additional delays and costs Li et al. (2022). In this context, synthetic data generation has emerged as a promising solution to address class imbalance and data scarcity, particularly for underrepresented classes. By augmenting training datasets with synthetic samples, models can achieve improved performance and generalization across different categories.

Synthetic data has gained popularity in recent years as a way to overcome the limitations of real-world data, which can be scarce, sensitive, or expensive to obtain (Patki et al. (2016)). Research in this area consistently highlights the potential of synthetic data to enhance the performance of ML models across diverse fields (Lu et al. (2023)), addressing challenges such as data shortages in computer vision and NLP (Mumuni et al. (2024)), generating diverse datasets in medical imaging (Frid-Adar et al. (2018b)), and providing safe training scenarios for autonomous driving systems (Song et al. (2023)). Its utility extends to financial modeling for algorithm testing under simulated market conditions and cybersecurity for developing threat detection systems (Potluru et al. (2023); Chalé & Bastian (2022)). In the domain of text analysis, synthetic data has been increasingly employed to enhance ML models, particularly in tasks such as text classification, sentiment analysis,

and natural language understanding. Moreover, researchers have shown generating synthetic samples with only targeted data examples could more effectively improve the model (Jin et al. (2024)). However, the process of identifying these optimal data examples often requires substantial domain expertise and manual effort, making it time-consuming and less scalable for real-world applications.

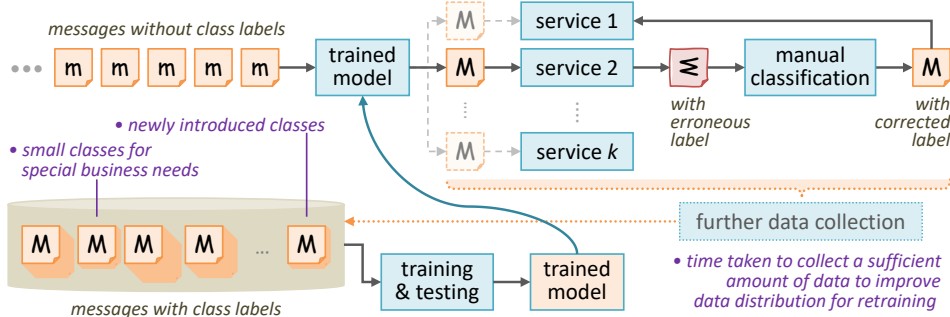

Figure 1: The workflow for developing and deploying a classification model in an industrial ticketing system, and the main obstacles impacting on the performance of the model.

To overcome these challenges, this paper introduces *Automated Generation of Text Synthetics* (AutoGeTS), an algorithmic solution that automates the search for optimal data examples based on specific improvement objectives, eliminating the need for human intervention. Through experiments with three search strategies and four objective functions, we identify key patterns between optimal strategy-objective combinations and data characteristics. We propose an ensemble algorithm that effectively improves text classification models across various real-world tasks.

## 2  RELATED WORK

Synthetic data is increasingly used as a powerful tool for generating realistic datasets to enhance the performance of the task across various domains (Meier et al. (1988); Bersano et al. (1997)). Early synthetic data generation methods include bootstrapping (Efron (1992); Breiman (1996)), which resamples from original data to estimate distributions and reduce variance in predictions, proved effective for a range of predictive algorithms including tree-based models (Sutton (2005)). However, bootstrapping couldn't introduce new patterns. The Synthetic Minority Over-sampling Technique (SMOTE) (Chawla et al. (2002)) advanced imbalanced dataset handling but risked overfitting. Data augmentation (Jaderberg et al. (2014)) improved model robustness by transforming existing data points, increasing diversity, yet still limited to patterns in the original dataset. The advent of deep learning introduced more sophisticated techniques, notably Generative Adversarial Networks (GANs) by Goodfellow et al. (Goodfellow et al. (2014)), which generate highly realistic synthetic data capturing dataset complexity. Studies have shown models trained on GAN-generated synthetic data often perform comparably to those trained on real data in various predictive tasks (Zhang et al. (2017); Cortés et al. (2020)). Frid-Adar et al. (Frid-Adar et al. (2018a)) enhanced liver lesion diagnosis using GAN-generated images, while Yale et al. (2020) demonstrated comparable performance using GAN-generated synthetic electronic health records for ICU patient predictions.

GANs have been extensively used for synthetic text generation. For instance, Croce et al. (2020) demonstrated their effectiveness in generating realistic text for NLP tasks, while He et al. (2022) explored task-specific text generation. However, GAN-generated data for text classification often lacks semantic coherence and relevance to specific tasks (Torres (2018)). Recent advancements in large language models (LLMs), such as GPT-2 (Croce et al. (2020)), provide new approaches to overcome these limitations. LLMs excel in few-shot and zero-shot learning (Brown (2020); Wang et al. (2021)), adapting to unseen tasks and generating contextually relevant data that improves model robustness. Yoo et al.'s GPT-3Mix (Yoo et al. (2021)) demonstrates LLMs' capability to generate diverse, high-quality synthetic data for text classification through careful prompt engineering. Prompt optimization strategies have shown that carefully crafting input prompts can significantly impact the quality of generated data (Wang et al. (2023)). Automated search techniques for identifying the most effective prompts, such as those used in AutoPrompt (Shin et al. (2020); Xu et al. (2024)),

offer a potential solution for improving synthetic data generation. Beyond prompt engineering, selecting appropriate input examples has emerged as a crucial focus. Selecting example data, either with a uniform distribution or human identification through VIS4ML, to form the prompt for LLM to generate synthetics is shown effective (Li et al. (2023); Jin et al. (2024)). Despite these advancements, LLM-generated data still struggles to fully capture real-world diversity, especially in highly subjective tasks (Li et al. (2023)). To address this, we propose AutoGeTS, an automated approach that optimizes input example selection for LLM-generated synthetic data. Designed for real-world business requirements, AutoGeTS reduces human intervention while systematically identifying impactful examples, enhancing model performance in scenarios of data scarcity and class imbalance.

## 3 METHODS

### 3.1 AUTOGETS ARCHITECTURE AND WORKFLOW

Figure 2 illustrates the AutoGeTS architecture. After training and evaluating the original model $M0$, improvement requirements (overall or class-specific) are determined. For a selected class $C$, visual encoding is applied to the training dataset. The optimal strategy-objective is employed to select example message sets $E_s$ from $C$, which are then processed through GPT-3.5's API using a zero-shot prompt template, one message per chat (detailed in Appendix B.1). Each example generates multiple synthetic samples through automated parsing and format cleaning of the LLM responses. These samples are appended to the training set for model retraining. The best-performing model in testing, according to the specified goals, is selected for deployment.

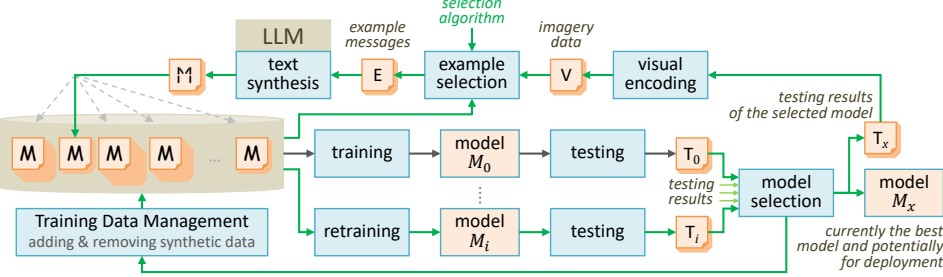

Figure 2: The architecture of AutoGeTS and the workflow for training and improving a model.

### 3.2 OBJECTIVES FOR MODEL OPTIMIZATION

Ticketing systems deployed in specific organizational environments often face different, sometimes conflicting, requirements. Typical business requirements and related performance metrics include:

R1. The accuracy of every class should be as high as possible and above a certain threshold. One may optimize a model with a performance metric such as class-based *balanced accuracy* or *F1-score* as the objective function, with each threshold value as a constraint.

R2. The overall classification accuracy of a model should be as high as possible and above a certain threshold because misclassified messages lead to undesirable consequences. One may optimize a model with a global performance metric, such as overall *balanced accuracy* and overall *f1-score*.

R3. The recall for some specific classes (e.g., important) should be as high as possible and above a certain threshold in order to minimize the delay due to the messages in such a class being sent to other services. Class-based *recall* is the obvious metric for this requirement. Often one may make a balanced judgment by observing Pareto fronts of *recall* in conjunction with another class-based metric (e.g., *balanced accuracy* or *F1-score*).

These requirements inform the definition of objective functions and constraints for AutoGeTS optimization. However, because the use of LLMs to generate synthetic data to aid ML (i.e., the workflow in Figure 2) is a recent approach, it is necessary to understand how different example selection algorithms for LLMs may impact the optimization.

### 3.3 STRATEGIES FOR EXAMPLE SELECTION

Defining the search space for example selection is critical, especially when augmenting datasets with synthetic examples. This space includes all possible subsets of training data $D = x_1, x_2, \ldots, x_n$, each data dot labeled with a specific class. The primary objective is to identify the optimal subset $W^* \subseteq D$ that maximizes performance metrics when used for synthetic data generation via LLM. With $2^n - 1$ possible subsets for $n$ examples, exhaustive search becomes intractable, necessitating heuristic strategies.

The general goal can be formulated as an ideal multi-objective optimization problem:

$$W^* = \arg \max_{W \subseteq D} J(W) \tag{1}$$

where $J(W)$ is the objective function measuring the performance of a retrained model $M(W)$ using synthetic data generated from subset $W$:

$$J(W) = w_1 \cdot \text{Recall}(M(W)) + w_2 \cdot \text{BalancedAccuracy}(M(W)) + w_3 \cdot \text{F1}(M(W)) \tag{2}$$

where $w_1$, $w_2$, and $w_3$ reflect metrics weights in the overall objective. Given the practical challenges in defining such a compound objective, we employ a simplified, single-metric function:

$$W^* = \arg \max_{W \subseteq D} J'(W) \tag{3}$$

where $J'(W)$ represents one of the following metrics: Class-based Recall (CR), Class-based Balanced Accuracy (CBA), Overall Balanced Accuracy (OBA), and Overall F1-Score (OF1).

Thus, the policy for selecting optimal subsets involves two core components:

1. A strategy for selecting subsets of examples for synthetic data generation and retraining.
2. An evaluation metric, $J'(W)$, to be maximized as the objective for the search towards the optimal subset $W^*$.

The challenge is to efficiently search the space while balancing between computational cost (i.e., cumulative model retraining time) and performance improvement (i.e., maximum gain in $J(W)$). Initial experiments with random selection yielded limited improvements, particularly for class-based metrics (detailed in Appendix B.1.3). Thus, we explore three primary strategies to optimize the subset selection: brute-force (Sliding Window, SW), gradient-based (Hierarchical Sliding Window, HSW), and evolutionary algorithms (Genetic Algorithm, GA), as illustrated in Figure 3.

### 3.3.1 SLIDING WINDOW (SW)

The Sliding Window (SW) strategy represents a brute-force approach, where the search space is exhaustively segmented into "windows" or subsets. For each window $W_k \subseteq D$, synthetic data is generated, the model is retrained, and the performance is evaluated based on the objective function $J'(W_k)$. The goal is to identify the window $W_k^*$ that yields the maximum improvement:

$$W_k^* = \arg \max_{W_k \subseteq D} J'(W_k) \tag{4}$$

The brute-force nature of SW ensures that no region of the search space is neglected, but the cost in terms of time and computational resources can become prohibitive.

### 3.3.2 HIERARCHICAL SLIDING WINDOW (HSW)

The Hierarchical Sliding Window (HSW) strategy builds on the principles of hierarchical selection, offering a more computationally efficient approach by incrementally narrowing the search space to promising regions. At each level $l$, the current search space is partitioned into smaller windows $W_{k,l}$. For each window, synthetic data is generated, the model is retrained, and the performance

is evaluated. Only the windows with the highest objective function values are selected for further hierarchical subdivision in the next level:

$$W_{k,l+1}^* = \arg \max_{W_{k,l+1} \in \text{Subspace}(W_{k,l})} J'(W_{k,l+1}) \tag{5}$$

The process repeats until improvement in $J'(W)$ plateaus or a predefined stopping criterion is met. HSW thus is akin to a targeted optimization approach that progressively homes in on the optimal subset $W^*$, balancing thorough exploration with reduced computational complexity compared to the brute-force SW method.

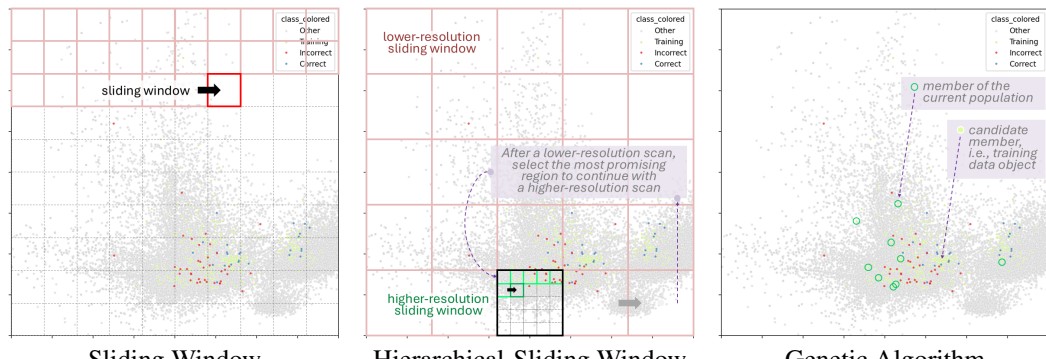

| Sliding Window | Hierarchical-Sliding Window | Genetic Algorithm |

Figure 3: The three examples subset selection strategies.

### 3.3.3 GENETIC ALGORITHM (GA)

The Genetic Algorithm (GA) begins by initializing a population

$$P = \{S_1, S_2, \ldots, S_m\} \tag{6}$$

where each candidate solution $S_i \subseteq D$ represents a subset of the training data $D$, encoded as a priority value-based chromosome. Each above threshold element indicates that the corresponding data example is included in the subset.

The GA evolves this population over generations, guided by a fitness score $F(S)$ defined as the objective function $J'(S)$, derived from the AutoGeTS process and subsequent performance evaluation of the retrained model $M(S)$. The algorithm applies three main genetic operators: **Selection**: At each generation, Lexicase selection Spector (2012) and Clustered Tournament selection Xie & Zhang (2012) are employed to select individuals into the mating pool based on their fitness, where Lexicase selection evaluates the $F(S)$ of input class and the $J'(S)$ of other randomly chosen classes. **Crossover**: Weight Mapping Crossover Gen et al. (2006) is used to combine two parent solutions $S_i$ and $S_j$ from the mating pool to produce offspring $O_k$ for local exploration. **Mutation**: Adaptive Polynomia Mutation Si et al. (2011) is applied to offspring $O_k$ to introduce variability for global search. The GA repeats the selection, crossover, and mutation process until reaching a specified number of generations or a convergence criterion. The subset $S^*$ that maximizes the fitness score:

$$S^* = \arg \max_{S_i \in P} F(S_i) \tag{7}$$

where $F(S_i) = J'(S_i)$, is finally retrieved. For further details and the step-by-step breakdown of HSW and GA algorithms, refer to Algorithm 1 and 2 in Appendix C.1.

Each AutoGeTS run targets a single class $C_i$, aiming to improve its specific or overall performance through synthetic sample addition. However, class interactions in synthetic data generation have been observed; Jin et al. (2024) found that synthetic data for one class can improve performance for others. Given these interactions and the collective contribution of all classes to overall classification

performance, an ensemble algorithm applying AutoGeTS across multiple classes is necessary to optimize both class-specific and overall performance.

## 3.4 ENSEMBLE ALGORITHM

The ensemble algorithm depends on the specific business requirements, as outlined in Section 3.2:

**To lift all classes performances above a threshold (R1)**: Iteratively apply AutoGeTS to each underperforming class ($C_{low}$) with optimal strategy-objective combination, in the order that most likely improves class performance. In each iteration, append synthetic samples from the optimal retrained model to the training set, and maintain improvements for processed $C_{low}$ above a specified threshold. Terminate when unable to maintain improvements. **To improve overall classification accuracy (R2)**: The same process is applied to each class ($C$), except the class order that most likely improves overall performance is used, and the algorithm terminates when overall performance plateaus. **To improve specific important class's performance (R3)**: For an important class ($IC$), identify related classes ($RC$) that could enhance $IC$ performance with AutoGeTS. Apply AutoGeTS to $IC$ and $RC$ iteratively with optimal strategy-objective combinations, in the order that prioritizes $IC$ performance improvement. Terminate when $IC$ performance plateaus.

Experimental determinations include performance thresholds, $RC$ identification, class order, and optimal strategy-objective combinations, which will be studied in Section 4. The specific details and the step-by-step breakdown of the algorithms are provided in Algorithm 3, 4, and 5 in Appendix C.2.

## 4 EXPERIMENTS AND RESULTS

AutoGeTS is evaluated through 1 GPU hour fixed-time experiments to improve $M0$ in meeting business requirements and to determine optimal strategy-objective policy as outlined in Section 3.2.

### 4.1 EXPERIMENT SETUP

Table 1: Original CatBoost Model $M0$ Performance

| Class | Class Size | Balanced Accuracy | Recall | F1-Score |
|---|---|---|---|---|
| T2 | 11350 | 0.950 | 0.941 | 0.921 |
| T1 | 8529 | 0.986 | 0.979 | 0.977 |
| T3 | 4719 | 0.952 | 0.914 | 0.922 |
| T5 | 2755 | 0.889 | 0.794 | 0.794 |
| T7 | 1963 | 0.883 | 0.780 | 0.766 |
| T6 | 1888 | 0.821 | 0.665 | 0.623 |
| T10 | 1699 | 0.761 | 0.540 | 0.554 |
| T9 | 1466 | 0.861 | 0.747 | 0.680 |
| T4 | 1387 | 0.899 | 0.801 | 0.859 |
| T8 | 1028 | 0.828 | 0.665 | 0.672 |
| T14 | 764 | 0.772 | 0.548 | 0.607 |
| T15 | 543 | 0.726 | 0.452 | 0.596 |
| T11 | 471 | 0.973 | 0.947 | 0.967 |
| T12 | 358 | 0.742 | 0.484 | 0.608 |
| T13 | 180 | 0.666 | 0.333 | 0.469 |
| Overall | 39100 | 0.923 | 0.856 | 0.856 |

We evaluated the AutoGeTS framework using a dataset from an enterprise IT support ticketing system, comprising 39,100 entries labeled into 15 task classes. The dataset is highly imbalanced, with some classes representing less than 1% of the total entries. To mitigate the effect of this imbalance, we split the dataset into 80% for training/validation and 20% for testing, with a further 80-20 split on the training set for validation. The imbalanced nature of the dataset mirrors real-world challenges faced by classification systems in industrial applications.

We used GPT-3.5 (version: 2023-03-15-preview) to generate synthetic text, employing parameters such as temperature = 0.7, max tokens = 550, top p = 0.5, frequency penalty = 0.3, and presence penalty = 0.0. Comparative experiments with the Easy Data Augmentation (EDA) tool (Wei & Zou (2019)), a traditional data augmentation method, demonstrated that while AutoGeTS improved

performance with both approaches, LLM-based workflow yielded superior results (detailed in Appendix E). For the baseline classification model, we utilized CatBoost with fixed hyperparameters (300 iterations, learning rate = 0.2, depth = 8, L2 leaf regularization = 1) to ensure consistency across all retrained models. More detailed $M0$ analysis and prompt are provided in Appendix A and B.1. The effectiveness of AutoGeTS was evaluated using class-based balanced accuracy, recall, and F1-score for local performance, as well as overall balanced accuracy and F1-score for global performance. The performance of the original CatBoost model $M0$ is shown in Table 1.

Table 2: Performance Comparison with M0, comparing Overall and Class Balanced Accuracy.

| Class Name | Class Size | M0 Bal Acc | Sliding Window Overall | Sliding Window Class | Hierarchical SW Overall | Hierarchical SW Class | Genetic Algorithm Overall | Genetic Algorithm Class |
|---|---|---|---|---|---|---|---|---|
| T2 | *11350* | 0.950 | ▲0.0030 | ▲0.0050 | ▲0.0034 | ▲0.0048 | ▲0.0009 | ▼0.0010 |
| T1 | *8529* | 0.986 | ▲0.0028 | ▲0.0005 | ▲0.0029 | ▲0.0005 | ▲0.0018 | ▼0.0018 |
| T3 | *4719* | 0.952 | ▲0.0030 | ▲0.0058 | ▲0.0027 | ▲0.0062 | ▲0.0029 | ▲0.0069 |
| T5 | *2755* | 0.889 | ▲0.0032 | ▲0.0189 | ▲0.0034 | ▲0.0140 | ▲0.0010 | ▲0.0059 |
| T7 | *1963* | 0.883 | ▲0.0036 | ▲0.0228 | ▲0.0034 | ▲0.0226 | ▲0.0012 | ▼0.0026 |
| T6 | *1888* | 0.821 | ▲0.0035 | ▲0.0190 | ▲0.0030 | ▲0.0196 | ▲0.0015 | ▲0.0073 |
| T10 | *1699* | 0.761 | ▲0.0034 | ▲0.0281 | ▲0.0044 | ▲0.0247 | ▲0.0027 | ▲0.0208 |
| T9 | *1466* | 0.861 | ▲0.0036 | ▲0.0147 | ▲0.0027 | ▲0.0191 | ▲0.0026 | ▲0.0077 |
| T4 | *1387* | 0.899 | ▲0.0029 | ▲0.0304 | ▲0.0033 | ▲0.0369 | ▲0.0036 | ▲0.0323 |
| T8 | *1028* | 0.828 | ▲0.0030 | ▲0.0321 | ▲0.0029 | ▲0.0358 | ▲0.0020 | ▲0.0142 |
| T14 | *764* | 0.772 | ▲0.0023 | ▲0.0326 | ▲0.0029 | ▲0.0395 | ▲0.0019 | ▲0.0396 |
| T15 | *543* | 0.726 | ▲0.0033 | ▲0.0456 | ▲0.0034 | ▲0.0446 | ▲0.0037 | ▲0.0533 |
| T11 | *471* | 0.973 | ▲0.0030 | ▲0.0054 | ▲0.0030 | ▲0.0053 | ▲0.0039 | ▲0.0053 |
| T12 | *358* | 0.742 | ▲0.0037 | ▲0.0699 | ▲0.0032 | ▲0.0772 | ▲0.0036 | ▲0.0775 |
| T13 | *180* | 0.666 | ▲0.0030 | ▲0.0443 | ▲0.0037 | ▲0.0548 | ▲0.0034 | ▲0.0548 |

## 4.2 PERFORMANCE IMPROVEMENTS OVERVIEW

The AutoGeTS framework yielded significant improvements in both local and global performance metrics, effectively addressing the class imbalance problem evident in Table 2.

Smaller, underrepresented classes experienced the largest improvements. For instance, T13's balanced accuracy increased by 5.48 percentage points (pp) from 66.6% in M0, while T12 showed a 7.75 pp improvement from 74.2%. In contrast, larger classes like T1 (98.6%) and T2 (95%) saw only marginal gains of around 0.3 pp. This demonstrates AutoGeTS's ability to significantly improve underrepresented classes without affecting the performance of well-represented ones. This balance is crucial in maintaining overall system performance. The overall balanced accuracy improved consistently and comparably among classes, with T10 showing the highest overall balanced accuracy improvement of 0.44 pp. This underscores AutoGeTS's synergistic effect, where class-based improvements translate to overall performance gains, with minimal trade-offs between local and global performance improvements (see Appendix D.1).

## 4.3 COMPARISON OF EXAMPLE SELECTION STRATEGIES AND OPTIMIZATION OBJECTIVES

Figure 4 compared the performance of three search strategies—SW, HSW, GA—and four objectives—maximizing CR, CBA, OBA, OF1—with respect to both local (class-specific) and global (overall) metrics. Each bar chart is divided into 4 sections for 4 performance metrics, with the four bars each representing the maximum improvement in the objective of maximizing CR, CBA, OBA, or OF1. The choice of strategy-objective played a critical role in the effectiveness of AutoGeTS, with each demonstrating distinct advantages depending on the size of the target class, and their performance trajectories over retraining time (see Appendix D.2).

For larger classes, HSW consistently yielded the best results, such as the highest class T1 balanced accuracy improvement of 0.5%. HSW's progressive narrowing of the search space proves effective for larger data sets where an exhaustive search is computationally prohibitive. Objective-wise, maximizing CR or CBA each best improved its respective metric, while maximizing OBA or OF1 both led to the best improvements in global metrics.

GA strategy proved superior for smaller and mid-sized classes, as shown by T13 and T12's highest balanced accuracy gains. GA's evolutionary nature generates diverse synthetic samples, crucial for small data sets. For these classes, maximizing CBA outperformed other objectives in local metrics, while OBA or OF1 maximization equally improved global metrics, except for T11, T12, and T15.

SW showed moderate performance across mid-sized classes, such as improving T5 and T6 balanced accuracy by up to 1.89% and 1.90%, respectively. SW offers a balanced trade-off between computational cost and performance improvement for mid-sized data sets. For these classes, maximizing CR or CBA equally improved both local metrics, and the same applies to maximizing OBA or OF1.

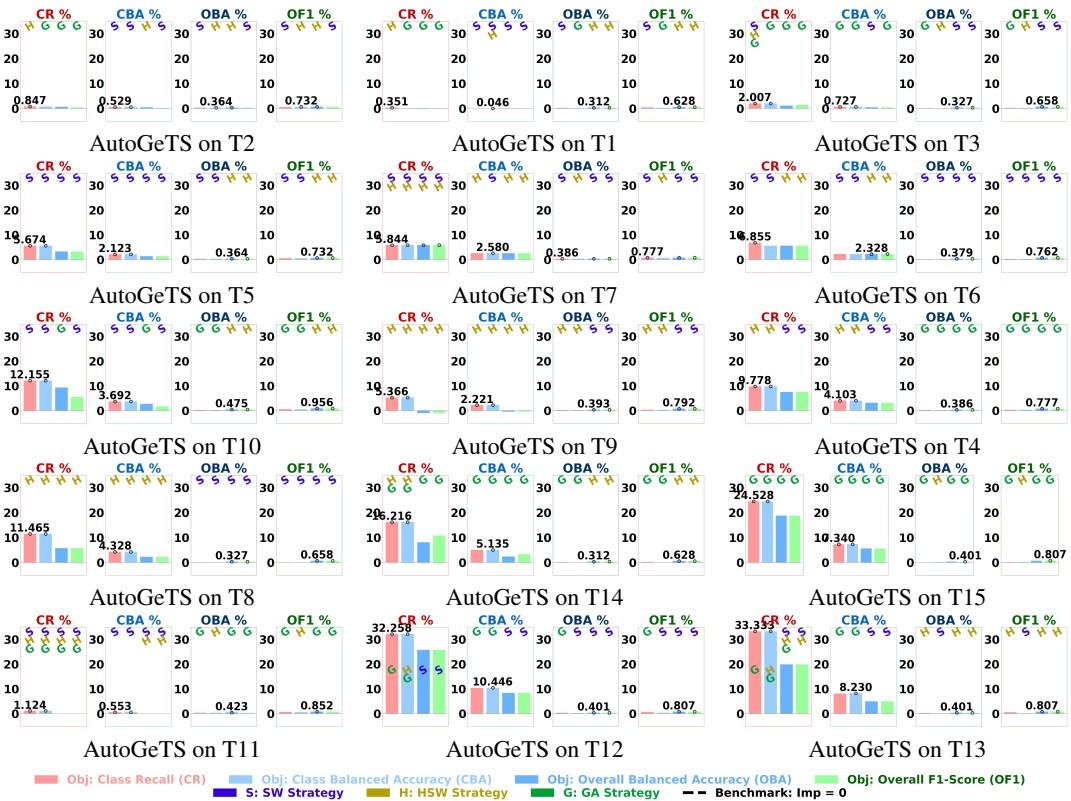

Figure 4: Comparison of improvements across 4 metrics for all classes, showing best-performing strategies (SW, HSW, GA) and highest improvement values for each objective

## 4.4 PARETO ANALYSIS FOR REPRESENTATIVE CLASSES

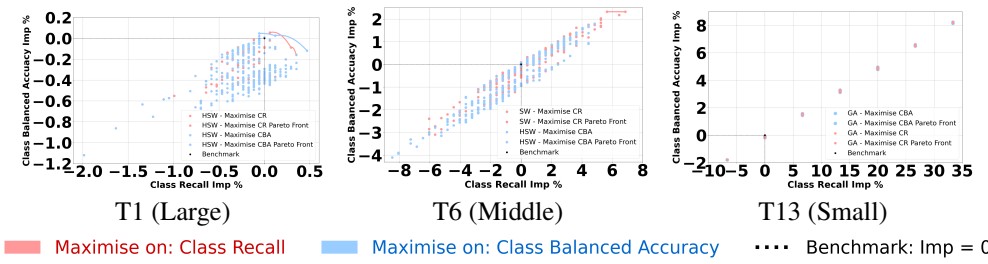

Figure 5: Improvements and Pareto Fronts in Class Recall vs Class Balanced Accuracy for topics T1, T6, and T13, showing models maximizing TR and TBA (note different axis ranges).

Figure 5 shows trade-offs between Class Recall (CR) and Class Balanced Accuracy (CBA) improvements for classes T1 (large), T6 (mid-sized), and T13 (small) when maximizing either CR or CBA.

CR improvements generally correlate with CBA gains, varying by class size. Large class T1 shows the largest divergence (CR +3.5%, CBA +1.2%). The more synthetic samples needed to impact a large class directly affects recall but indirectly specificity, leading to larger CR and CBA divergence. Mid-sized class T6 demonstrates aligned improvements, with targeting CR increasing CBA

by 2.9%, indicating a less critical objective choice between maximizing CR or CBA. Small class T13 exhibits substantial improvements in both CR and CBA regardless of which metric is maximized, as maximizing CR improved recall by 33.3% and CBA by 8.2%, reflecting the effectiveness of diverse synthetic samples and the GA strategy for underrepresented classes and small, imbalanced datasets. Practically, maximizing CR is preferable for large important classes to minimize misclassification delay, while either CR or CBA optimization can be effective for middle and small classes.

## 5 ENSEMBLE ALGORITHM AND FURTHER EXPERIMENTATION

### 5.1 SUMMARY OF STRATEGY-OBJECTIVE COMBINATIONS

Building on our analysis in Section 4.3, we synthesize the effectiveness of different strategy-objective combinations across varying class sizes and performance metrics.

Table 3: Optimal Strategy-Objective Combinations across Classes

| | Class Performance | | Overall Performance | |
|---|---|---|---|---|
| | Topic Recall | Topic Balanced Accuracy | Overall Balanced Accuracy | Overall F1-Score |
| T2 | HSW-TR | HSW-TBA/OBA | HSW-TBA/OBA | |
| T1 | | HSW-TBA | HSW-OBA/OF1 | |
| T3 | GA-TR/TBA | | SW-OBA/OF1 | |
| T5 | SW-TR/TBA | | HSW-OBA/OF1 | |
| T7 | SW-TBA | | SW-OBA/OF1 | |
| T6 | SW-TR | HSW-OBA/OF1 | | |
| T10 | SW-TR/TBA | | HSW-OBA/OF1 | |
| T9 | HSW-TR/TBA | | SW-OBA/OF1 | |
| T4 | | | GA-OBA/OF1 | |
| T8 | | | SW-OBA/OF1 | |
| T14 | GA-TBA | | HSW-OBA/OF1 | |
| T15 | | | GA-OF1 | |
| T11 | | | GA-OBA | |
| T12 | | | GA-OF1 | |
| T13 | | | HSW-OBA/OF1 | |

Table 3 summarizes optimal Strategy-Objective combinations for improving 4 metrics across all classes, serving as a look-up table for the ensemble algorithm in Section 5.

### 5.2 LOCAL AND GLOBAL METRICS IMPROVEMENT ACROSS CLASSES

We now examine the broader impacts of applying AutoGeTS to individual classes, considering inter-class effects and overall system performance. Figure 6 illustrates inter-class interaction and overall effects, guiding class order determination for three requirements: R1 (improve each class) orders classes by descending diagonal elements of class balanced accuracy improvements; R2 (improve overall performance) sorts classes by their overall balanced accuracy improvement (Figure 6b); and R3 (improve an important class) orders related classes $RC$ by their improvement on the class balanced accuracy of important class $IC$, according to the $IC$'s column of Figure 6a. The determined class orders for the three requirements are detailed in Appendix C.2.

### 5.3 ENSEMBLE CASE STUDY RESULTS FOR R3

We applied ensemble AutoGeTS to improve important class T13 following Algorithm 5 in Appendix C.2, with related classes T12, T10, T11, and T5 identified and ordered based on T13's column of Figure 6a. We selected optimal strategy-objective combinations for each class from Table 3. Benchmarks included iterating single strategies and random combinations. We conducted three runs with different train-validation splits and random seeds. T13 balanced accuracy was used as the performance metric instead of T13 recall, as they align well for T13 while reflecting changes in other classes (negative instances for T13) not captured by the recall.

The Ensemble Algorithm in Figure 7 achieved the highest T13 balanced accuracy and second-highest global improvements, with faster and consistently higher T13 performance gains. While

all methods are efficient, with HSW and GA achieving peak performance at 20% of retraining time and SW and ensemble at 40%, these results indicate a ceiling for iterative AutoGeTS enhancement.

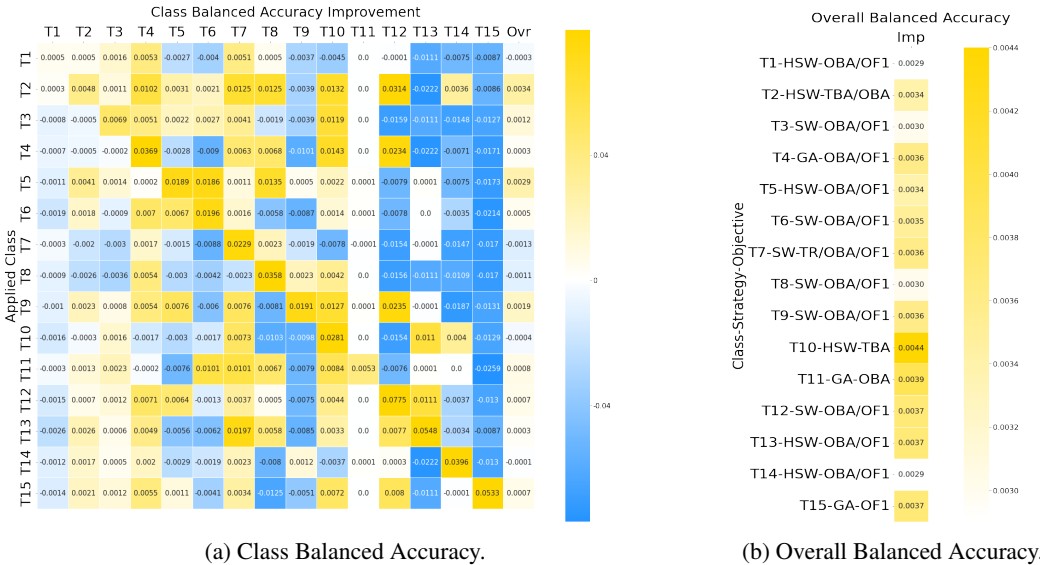

(a) Class Balanced Accuracy.

(b) Overall Balanced Accuracy.

Figure 6: Each class impact on class-based 6a and overall balanced accuracy 6b applying AutoGeTS.

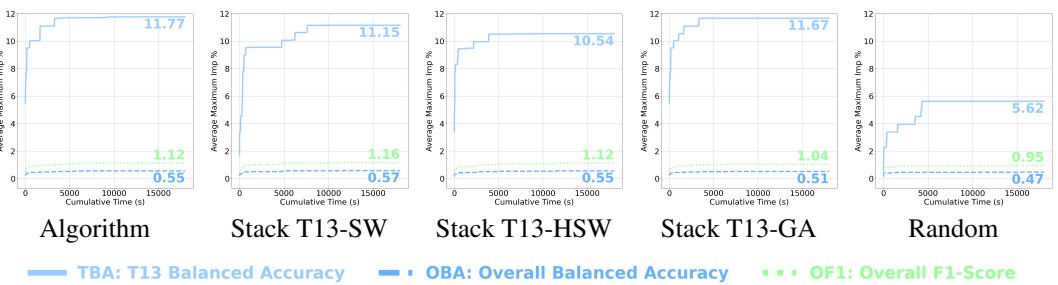

Figure 7: Average maximum cumulative T13 and overall improvements for 5 ensemble sequences.

## 6 CONCLUSIONS

This work introduces AutoGeTS, an automated framework optimizing example data selection for synthetic text generation using Large Language Models (LLMs). AutoGeTS significantly enhances the level of automation, reducing human efforts in selecting effective examples. This approach addresses class imbalance and data scarcity challenges in real-world text classification tasks. Experiments demonstrate AutoGeTS's effectiveness in improving both local and global performance metrics. Using Sliding Window, Hierarchical Sliding Window, and Genetic Algorithm, significant improvements in inadequately-sampled classes are observed without compromising well-represented ones. The ensemble algorithm for selecting the most suitable strategies according to the results of earlier iterations facilitates more efficient optimization processes in later iterations. This approach that treats earlier testing results as useful knowledge in optimization will likely have a wider application in ML model deployment. These findings establish AutoGeTS as an effective solution for enriching training data with synthetic samples to meet real-world requirements for the performance of ML models with limitations in data collection. This work confirms that the automated approach can perform better than human-centric processes in terms of both effectiveness and efficiency. Meanwhile, there is a need to conduct more large-scale experiments and analyze the experiment results in order to understand and explain how different synthetic data sets improve or undermine an ML model. Future work may also explore multi-objective optimization strategies, more advanced ensemble algorithms, and applications of this approach in other ML domains.

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

## APPENDICES

In the following appendices, we provide further experiment results through visualization plots. The experimental data will be made available on GitHub after the double-blind review process. These appendices include:

A. **Parameters for ML Training** — We report the pilot experiments for selecting parameters that were used in the training of the benchmark model M0 (trained without synthetic data). The relatively optimal parameter set was chosen and used for retraining all other models (trained with both collected data and synthetic data).

B. **Parameters for Example Search** — We report the pilot experiments selecting parameters to be used by different algorithms that search message examples to be used as the inputs to LLMs in order to generate synthetic data.

C. **Algorithms** — We report the detailed algorithm flowcharts for example data subset selection, determined after pilot experiments, and multi-class ensemble algorithms, determined through experiments in Section 4.

D. **Fixed-Time Experiments** — We report a set of experiments, where three workflows were allowed to use exactly one hour of GPU time for searching message examples, generating synthetic data, training and testing a model in multiple iterations in order to develop a model to improve the benchmark model M0.

E. **Comparison with Traditional Data Augmentation** — We provide comparative analysis between EDA-based and LLM-based AutoGeTS workflow, examining both the best improvements (overall and class-specific) and the temporal progression of overall balanced accuracy improvement across all 15 classes. The Easy Data Augmentation (EDA) tool (Wei & Zou (2019)) is a traditional data augmentation method.

## A    PILOT EXPERIMENTS: PARAMETER FOR ML TRAINING

Before developing the AutoGeTS framework, we aimed to improve the original CatBoost model, $M0$, through parameter tuning. A grid search was conducted with the following parameter ranges:

- learning_rate: [0.01, 0.05, 0.1, 0.2, 0.5]

- depth: [4, 6, 8, 10]

- l2_leaf_reg: [1, 3, 5, 10]

Five-fold cross-validation was employed, with overall classification accuracy as the primary criterion for evaluating the model performance.

## A.1 BENCHMARK M0 PARAMETERS EXPERIMENTS

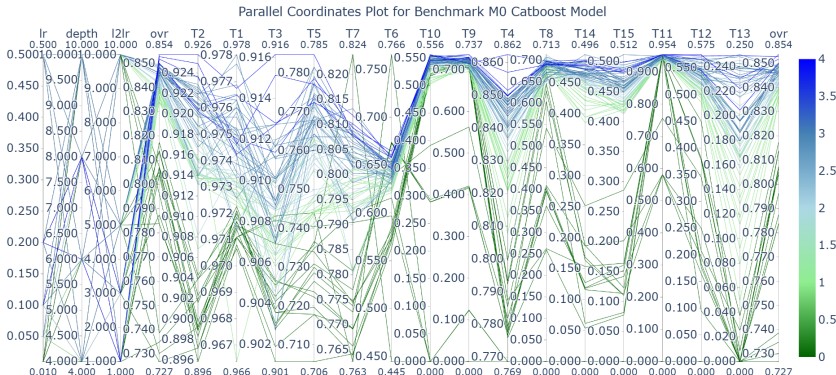

Figure 8: M0 Parameter Experiments Overview.

Figure 8 presents an overview of all experimented M0 CatBoost model parameters in a parallel coordinate plot. Each line represents a unique parameter set and its corresponding classification performance. The first three coordinates depict the experimented parameters: learning rate, tree depth, and L2 leaf regularization. The subsequent 15 coordinates (T1 to T15) represent the recall for each of the 15 classes, while the final coordinate shows the overall classification accuracy, which is the primary performance metric.

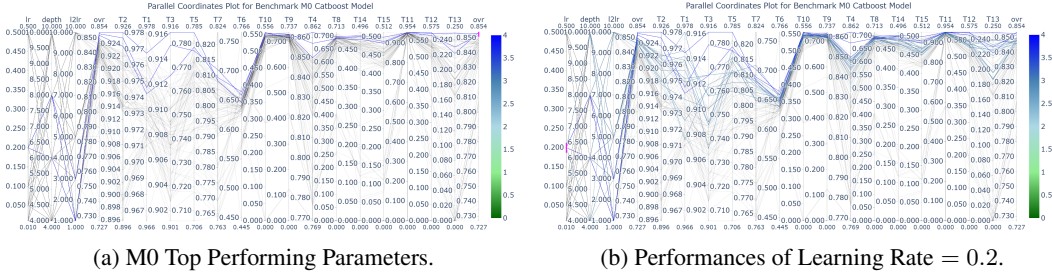

(a) M0 Top Performing Parameters.          (b) Performances of Learning Rate = 0.2.

Figure 9: M0 Top Performances and Learning Rate = 0.2.

Figure 9a highlights the top-performing parameter sets. These sets consistently use a learning rate of 0.2, tree depths of 6 or 8, and L2 leaf regularization values of 1 or 3. Based on these observations, we further examine the performance of learning_rate = 0.2 and determine the optimal values for depth and L2 leaf regularization.

Figure 9b highlights parameter sets with learning_rate = 0.2. All highlighted sets demonstrate good accuracy, including the three best accuracy scores, confirming 0.2 as the optimal learning rate among the experimented values.

Figures 10a and 10b compare model performances between depth = 6 and depth = 8 with learning rate set to 0.2. While both depth values yield good accuracy, depth = 8 shows slightly superior results overall.

Figures 11a and 11b compare model performances between L2_leaf_regularization = 1 and L2_leaf_regularization = 3 with learning rate

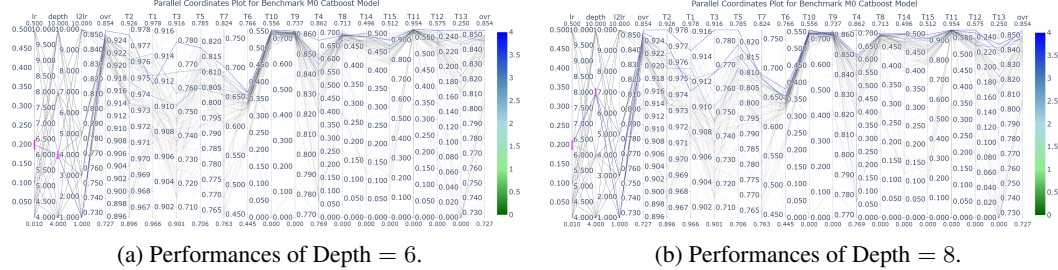

(a) Performances of Depth = 6.    (b) Performances of Depth = 8.

Figure 10: M0 Best Experimented Depth Value.

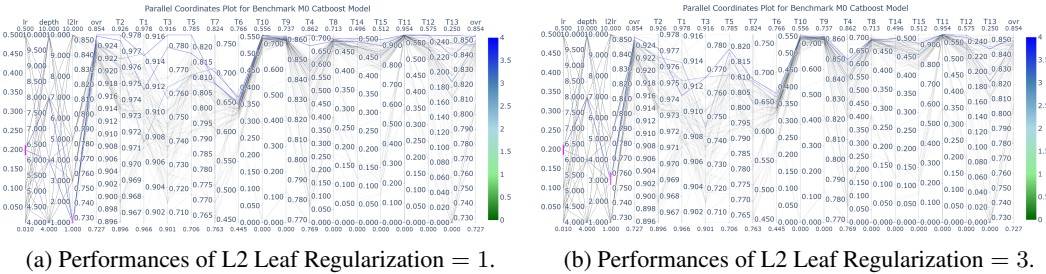

(a) Performances of L2 Leaf Regularization = 1.    (b) Performances of L2 Leaf Regularization = 3.

Figure 11: M0 Best Experimented L2 Leaf Regularization Value.

set to 0.2. Both values produce good performances, but L2_leaf_regularization = 1 demonstrates marginally better results.

Based on these analyses, the optimal parameter set for the M0 benchmark CatBoost model is:

- learning_rate = 0.2

- depth = 8

- L2_leaf_regularization = 1

This parameter set is used for all experiments involving the CatBoost model.

A.2 FURTHER ANALYSIS ON M0

Despite identifying a relatively optimal parameter set, analysis of class-specific performance revealed significant shortcomings. As shown in Table 1, several classes, particularly small or underrepresented ones (T12, T13, T14, and T15), exhibited unacceptable performance levels. These classes had balanced accuracies below 0.8 and recall rates around or below 0.5, potentially causing severe delays in messages and error reports processing.

To this end, we further realised that a serious class imbalance problem exists in the dataset with these small classes ranging only from 0.5% to 2% of the whole dataset, and 1.6% to 6.7% of the largest class. Moreover, data scarcity exists in these small classes, as illustrated in figure 12 that the red and blue dots distribute loosely across the plot. We therefore decided to investigate the use of synthetic data to improve this text classification model.

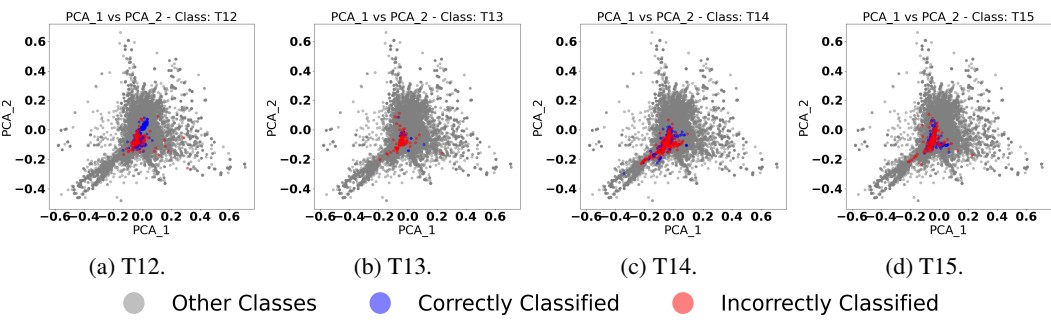

(a) T12.      (b) T13.      (c) T14.      (d) T15.

Other Classes    Correctly Classified    Incorrectly Classified

Figure 12: M0 PCA Plots for Small Classes T12, T13, T14, T15.

## B PILOT EXPERIMENTS: PARAMETERS FOR EXAMPLE SEARCH

In developing the AutoGeTS framework described in Section 3.1, we found that parameters for both the LLM's synthetic sample generation and the three example selection strategies significantly influenced AutoGeTS performance. To determine optimal parameter sets and understand their impact, we conducted extensive experiments on each component.

Given that our objectives for each retrained model were to maximize both overall accuracy and class-specific recall for the chosen class, we employed the Hypervolume (HV) indicator (Zitzler & Thiele (1999); Jiang et al. (2014)) to evaluate performance. This indicator allows us to compare results across different parameter configurations by considering both class-based recall and overall accuracy simultaneously.

We implemented 5-fold cross-validation throughout our experiments. In addition to the HV indicator, we tracked the best accuracy and best class recall across all five folds as supplementary performance metrics.

### B.1 SYNTHETIC DATA GENERATION PARAMETER EXPERIMENTS

The synthetic data generation process utilizes GPT-3.5 through its API interface. In preliminary experiments, we also evaluated Llama 3 as an alternative language model for synthetic sample generation, which yielded comparable results.

For each original text sample, we invoke a new chat session and employ a zero-shot approach without providing additional context. The input prompt template for generating synthetic samples follows this format:

```
'Generate ' + str(num) + ' lines of the data similar to this format data:
    + ' meta_data['text'].values[i] ' + 'put & at the end of each line'
```

where 'num' is the number of generated samples, meta_data is the input data, and 'i' is the data index number.

Upon receiving the LLM's response, we implement an automated pipeline for cleaning, parsing, and separating the output into 'num' synthetic samples based on the formatting parameters specified in the prompt template. A verification func-

tion inspects each synthetic sample for format quality assurance, checking includes:

- Empty samples or null responses
- Extraneous empty lines or spaces
- Correct placement of separation symbols ('&')

If 'm' samples fail these quality checks, the generation process is automatically repeated for the same input text, adjusting the prompt to request only the remaining 'm' samples (i.e., 'Generate ' + str(m) + ' lines...').

We investigated the impact of varying 'num' on AutoGeTS performance through a series of experiments.

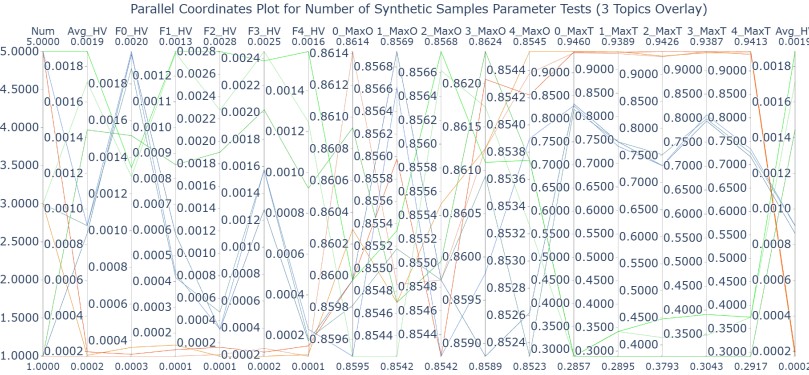

Figure 13: Synthetic Data Generation Parameter Experiments Overview (F0_HV: Fold 0 Hypervolume, MaxO: Max Overall Accuracy Improvement, MaxT: Max Topic Recall Improvement).

Figure 13 presents a parallel coordinate plot of synthetic text generation parameter experiments. The study utilized the Hierarchical Sliding Window approach, focusing on classes T2 (orange, largest class), T9 (blue, median size class), and T13 (green, smallest class). The primary parameter under investigation, "Syn Number," represents the number of synthetic text samples generated for each selected original text data point. The primary criterion, HV, appears as both the second-left and rightmost coordinates in the plot.

The results indicate that Syn Number = 5 consistently yielded the best Hypervolume for all three classes among the tested values. Consequently, we adopted the generation of five synthetic samples per selected original data point for all subsequent experiments.

### B.1.1 PCA PROJECTION OF SYNTHETIC SAMPLES

To verify the effectiveness of generated synthetic samples in addressing class imbalance and data scarcity, we projected these data using the same fitted vectorizer and PCA model used for the original data.

Figure 14 presents updated PCA plots for small classes T12, T13, T14, and T15. The inclusion of synthetic samples significantly increased the number of colored

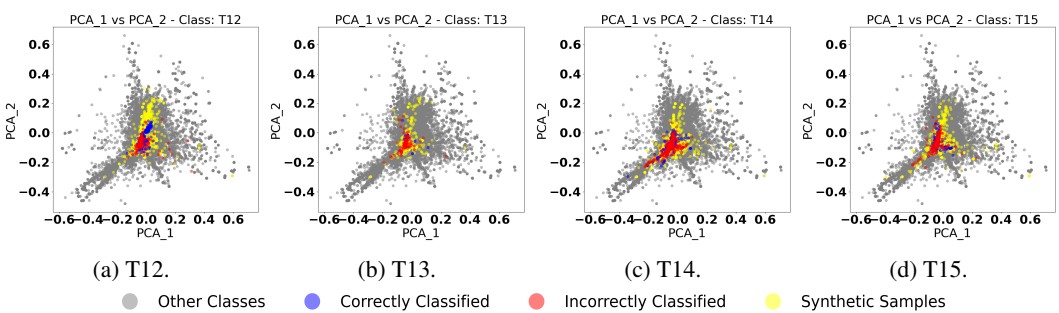

Figure 14: M0 PCA Plots with Synthetic Samples for Small Classes T12, T13, T14, T15.

data points. Moreover, these additional points appear more densely clustered, suggesting improvement on the previous data scarcity issues.

To illustrate the nature of the synthetic data generated, we present an example of an original data example and its corresponding LLM-generated synthetic samples. The following listing shows the original Spanish text followed by five synthetic samples, demonstrating how the LLM maintains the context and structure while introducing variations in content:

Listing 1: Original data example and LLM-generated synthetic samples

```
Original Data Example:
phone_nmb
En el ticket REQ0026231 se le solicita acceso a unidades de red pero
    sigue sin poder acceder y figura como resuelto, revisar por favor, la
     usuaria lleva 2 meses con este problema

Synthetic Samples:
1. El cliente reporta que su cuenta de correo electronico ha sido
    bloqueada, por favor revisar el caso REQ0027456.

2. Se solicita la instalacion de un software especifico en el equipo del
    usuario, el ticket es REQ0028745.

3. El usuario indica que no puede imprimir desde su equipo, se necesita
    revision del caso REQ0029367.

4. La usuaria reporta problemas con su conexion a internet, el ticket es
    REQ0030172.

5. Se requiere la asignacion de permisos adicionales en el sistema para
    el usuario, el caso es REQ0031298.
```

As evident from Listing 1, the synthetic samples maintain the overall structure of a ticketing system entry while diversifying the reported issues, demonstrating the LLM's ability to generate contextually relevant and varied data.

### B.1.2 SYNTHETIC SAMPLES PERFORMANCES WITHOUT EXAMPLES SELECTION

To evaluate the potential of our generated synthetic samples in improving the text classification model $M0$, we appended all generated synthetic samples to the training set and retrained the CatBoost model $M0$ for small classes T12, T13, T14, and T15.

Tables 4 and 5 present the results of this experiment. We observed that class-based performance often improved when synthetic samples for that class were appended, demonstrating the potential of synthetic data. However, only T12 showed improvement in overall performance. Notably, T13 failed to improve even its class-specific performance.

Table 4: Performance of Retrained Models with T12 and T13 Synthetic Samples

| Class | Δ Balanced Accuracy | | Δ Recall | | Δ F1-Score | |
|---|---|---|---|---|---|---|
| | T12 | T13 | T12 | T13 | T12 | T13 |
| T1 | ▼0.0014 | ▼0.0011 | ▼0.0034 | ▼0.0023 | ▼0.0006 | ▼0.0009 |
| T2 | ▲0.0036 | ▼0.0033 | ▲0.0044 | ▼0.0058 | ▲0.0053 | ▼0.0040 |
| T3 | ▲0.0015 | ▼0.0003 | ▲0.0032 | ▲0.0011 | ▲0.0013 | ▼0.0054 |
| T4 | ▲0.0104 | ▼0.0001 | ▲0.0214 | 0.0000 | ▲0.0064 | ▼0.0016 |
| T5 | ▼0.0015 | ▼0.0018 | ▼0.0038 | ▼0.0038 | ▲0.0022 | ▼0.0015 |
| T6 | ▼0.0021 | ▲0.0036 | ▼0.0027 | ▲0.0080 | ▼0.0102 | ▼0.0003 |
| T7 | ▲0.0023 | ▼0.0039 | ▲0.0025 | ▼0.0076 | ▲0.0161 | ▼0.0065 |
| T8 | ▲0.0046 | ▼0.0104 | ▲0.0085 | ▼0.0212 | ▲0.0144 | ▼0.0101 |
| T9 | ▼0.0011 | ▼0.0066 | ▼0.0036 | ▼0.0144 | ▲0.0093 | ▲0.0014 |
| T10 | ▲0.0094 | ▲0.0043 | ▲0.0179 | ▲0.0090 | ▲0.0192 | ▲0.0041 |
| T11 | ▲0.0001 | 0.0000 | 0.0000 | 0.0000 | ▲0.0053 | 0.0000 |
| T12 | ▲0.0376 | ▼0.0234 | ▲0.0781 | ▼0.0469 | ▼0.0497 | ▼0.0364 |
| T13 | ▼0.0333 | ▼0.0127 | ▼0.0667 | ▼0.0222 | ▼0.0753 | ▼0.1506 |
| T14 | ▲0.0110 | ▲0.0040 | ▲0.0222 | ▲0.0074 | ▲0.0144 | ▲0.0184 |
| T15 | ▼0.0045 | ▼0.0298 | ▼0.0085 | ▼0.0598 | ▼0.0177 | ▼0.0543 |
| Overall | ▲0.0015 | ▼0.0023 | ▲0.0028 | ▼0.0043 | ▲0.0028 | ▼0.0043 |

Table 5: Performance of Retrained Models with T14 and T15 Synthetic Samples

| Class | Δ Balanced Accuracy | | Δ Recall | | Δ F1-Score | |
|---|---|---|---|---|---|---|
| | T14 | T15 | T14 | T15 | T14 | T15 |
| T1 | ▼0.0011 | ▼0.0004 | ▼0.0029 | ▼0.0017 | ▼0.0003 | ▲0.0005 |
| T2 | ▼0.0014 | ▼0.0039 | ▼0.0071 | ▼0.0097 | ▲0.0011 | ▼0.0030 |
| T3 | ▼0.0027 | ▼0.0027 | ▼0.0054 | ▼0.0043 | ▼0.0029 | ▼0.0063 |
| T4 | ▼0.0071 | ▲0.0069 | ▼0.0142 | ▲0.0142 | ▼0.0071 | ▲0.0037 |
| T5 | ▼0.0035 | ▼0.0044 | ▼0.0075 | ▼0.0094 | ▼0.0023 | ▼0.0027 |
| T6 | ▲0.0023 | ▼0.0027 | ▲0.0054 | ▼0.0054 | ▼0.0012 | ▼0.0042 |
| T7 | ▲0.0073 | ▲0.0023 | ▲0.0152 | ▲0.0051 | ▲0.0053 | ▼0.0007 |
| T8 | ▼0.0080 | ▼0.0104 | ▼0.0169 | ▼0.0212 | ▼0.0013 | ▼0.0101 |
| T9 | ▼0.0024 | ▼0.0031 | ▼0.0072 | ▼0.0072 | ▲0.0165 | ▲0.0047 |
| T10 | ▼0.0095 | ▲0.0016 | ▼0.0179 | ▲0.0030 | ▼0.0200 | ▲0.0039 |
| T11 | 0.0000 | 0.0000 | 0.0000 | 0.0000 | 0.0000 | 0.0000 |
| T12 | ▼0.0156 | ▲0.0001 | ▼0.0313 | 0.0000 | ▼0.0278 | ▲0.0122 |
| T13 | 0.0000 | ▼0.0111 | 0.0000 | ▼0.0222 | 0.0000 | ▼0.0243 |
| T14 | ▲0.0135 | ▼0.0259 | ▲0.0370 | ▼0.0519 | ▼0.1219 | ▼0.0388 |
| T15 | ▼0.0128 | ▲0.0219 | ▼0.0256 | ▲0.0513 | ▼0.0241 | ▼0.1079 |
| Overall | ▼0.0026 | ▼0.0026 | ▼0.0049 | ▼0.0049 | ▼0.0049 | ▼0.0049 |

These mixed results suggest that indiscriminate use of all generated synthetic samples may not consistently yield improvements. This observation led us to conclude that a selective approach to choosing text examples for synthetic data generation is necessary. Such selection consequently filters the generated samples to be appended for retraining the text classification model.

To evaluate the effectiveness of selection strategies, we first established a random selection baseline, followed by our proposed strategic selection methods. The following sections present these evaluations, beginning with the random selection baseline results.

### B.1.3  SYNTHETIC SAMPLES PERFORMANCES WITH RANDOM EXAMPLES SELECTION

To establish a baseline for evaluating selection strategies, we implemented a random selection approach. For each target class, this baseline process randomly samples a random number (between 1 and the size of the class pool) of examples with replacement, where selected examples subsequently go through the synthetic data generation process described at the beginning of Appendix B.1. We evaluated this baseline using 1 GPU hour fixed-time experiments to improve M0, maintaining consistency with the experimental settings described in Section 4.

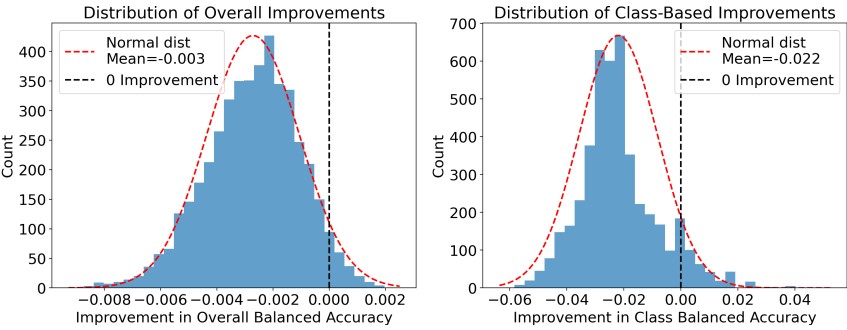

Figure 15: Random Selection Results Distribution on Improving OBA (left) and CBA (Right).

The optimization process ran multiple iterations for 1 GPU hour, averaging 299 iterations per class, with 4487 sets of random selected examples tested in total. Figure 15 presents the performance distribution of models retrained using each example set. The overall balanced accuracy (OBA) improvement ranged from -0.9% to 0.2%, with an average of -0.3% and only 3.9% of retraining leading to positive OBA improvement. At the class level, the class balanced accuracy (CBA) improvement fluctuated between -5.8% and 4.7%, with an average of -2.2% and 9.1% of retraining achieving positive CBA improvement.

Table 6: Performance Comparison between Random Selection and Strategic Selections.

| Class | Random | | Sliding Window | | Hierarchical SW | | Genetic Algorithm | |
| Name | Overall | Class | Overall | Class | Overall | Class | Overall | Class |
|---|---|---|---|---|---|---|---|---|
| T2 | ▲0.0012 | ▼0.0155 | ▲0.0030 | ▲0.0050 | ▲0.0034 | ▲0.0048 | ▲0.0009 | ▼0.0010 |
| T1 | ▲0.0009 | ▼0.0202 | ▲0.0028 | ▲0.0005 | ▲0.0029 | ▲0.0005 | ▲0.0018 | ▼0.0018 |
| T3 | ▲0.0014 | ▼0.0148 | ▲0.0030 | ▲0.0058 | ▲0.0027 | ▲0.0062 | ▲0.0029 | ▲0.0069 |
| T5 | ▲0.0002 | ▼0.0136 | ▲0.0032 | ▲0.0189 | ▲0.0034 | ▲0.0140 | ▲0.0010 | ▲0.0059 |
| T7 | ▲0.0018 | ▼0.0091 | ▲0.0036 | ▲0.0228 | ▲0.0034 | ▲0.0226 | ▲0.0012 | ▼0.0026 |
| T6 | ▲0.0016 | ▼0.0124 | ▲0.0035 | ▲0.0190 | ▲0.0030 | ▲0.0196 | ▲0.0015 | ▲0.0073 |
| T10 | ▼0.0006 | ▲0.0051 | ▲0.0034 | ▲0.0281 | ▲0.0044 | ▲0.0247 | ▲0.0027 | ▲0.0208 |
| T9 | ▲0.0016 | ▲0.0049 | ▲0.0036 | ▲0.0147 | ▲0.0027 | ▲0.0191 | ▲0.0026 | ▲0.0077 |
| T4 | ▲0.0018 | ▲0.0118 | ▲0.0029 | ▲0.0304 | ▲0.0033 | ▲0.0369 | ▲0.0036 | ▲0.0323 |
| T8 | ▲0.0012 | ▲0.0146 | ▲0.0030 | ▲0.0321 | ▲0.0029 | ▲0.0358 | ▲0.0020 | ▲0.0142 |
| T14 | ▲0.0009 | ▲0.0029 | ▲0.0023 | ▲0.0326 | ▲0.0029 | ▲0.0395 | ▲0.0019 | ▲0.0396 |
| T15 | ▲0.0005 | ▲0.0067 | ▲0.0033 | ▲0.0456 | ▲0.0034 | ▲0.0446 | ▲0.0037 | ▲0.0533 |
| T11 | ▲0.0020 | ▼0.0248 | ▲0.0030 | ▲0.0054 | ▲0.0030 | ▲0.0053 | ▲0.0039 | ▲0.0053 |
| T12 | ▲0.0014 | ▲0.0472 | ▲0.0037 | ▲0.0699 | ▲0.0032 | ▲0.0772 | ▲0.0036 | ▲0.0775 |
| T13 | ▲0.0006 | ▲0.0237 | ▲0.0030 | ▲0.0443 | ▲0.0037 | ▲0.0548 | ▲0.0034 | ▲0.0548 |

The best results achieved through random selection are presented in Table 6. While overall performance improvements were observed, the random selection struggled particularly with improving class performance for larger classes.

One set of plots showing the overall balanced accuracy (OBA) improvements over time for the 15 classes is presented in Figure 16, where the random selection benchmark results are shown in gray dash-lines. We can observe that random selection frequently achieves only marginal improvements over the original model, as evidenced in classes T7, T13, T14, and T15. Moreover, for T10, the random selection approach performed worse than the original model.

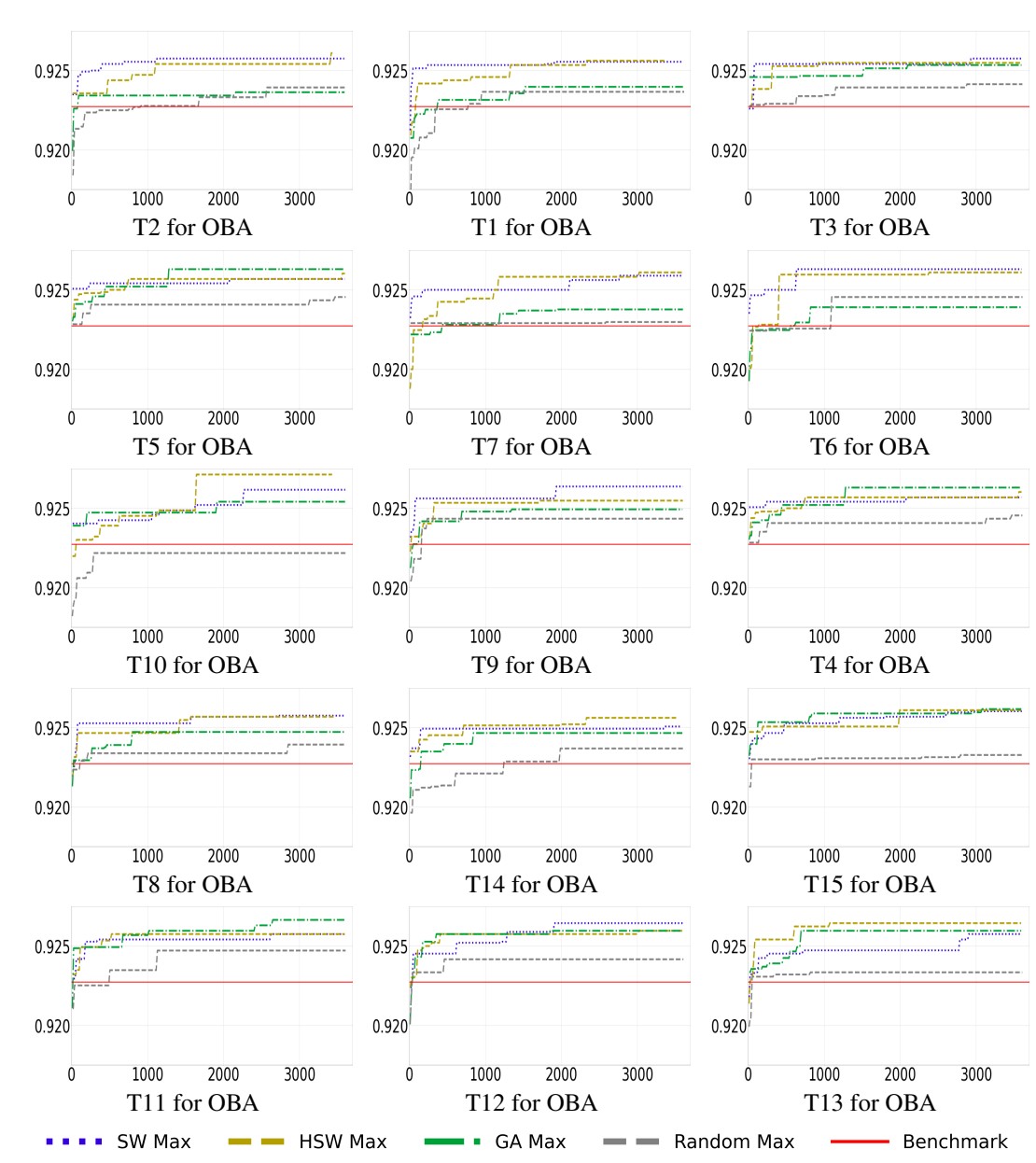

Figure 16: Fixed 1 Hour GPU time (x-axis, in seconds), Comparing on OBA Improvement (y-axis).

These findings motivated us to develop the three example selection strategies and conduct parameter studies for each, which we present in the following sections.

## B.2 EXAMPLE SELECTION STRATEGIES PARAMETER EXPERIMENTS

### B.2.1 SLIDING WINDOW (SW) PARAMETERS EXPERIMENTS

We investigated the parameters of the Sliding Window strategy. Figure 17a presents an overview of the experimental results. The parameters under investigation are:

- Area Size (AS): the size of the area to which the sliding window is applied
- Num Seg (NS): the number of segments/bins per dimension
- Window Size (WS): the number of bins per window

These parameters are represented by the three leftmost coordinates in the plot. We used the Hypervolume Indicator as the primary comparison metric, supplemented by the maximum values of both objectives in each of the 5 cross-validation folds.

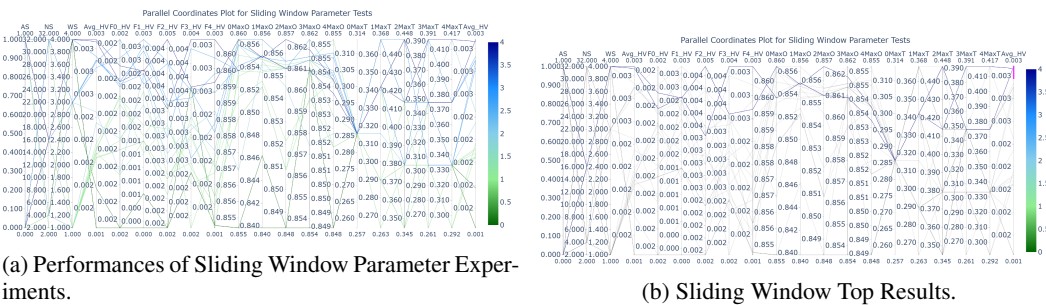

(a) Performances of Sliding Window Parameter Experiments.

(b) Sliding Window Top Results.

Figure 17: Sliding Window Parameter Experiments Overview and Top Results.

Figure 17b highlights the top-performing parameter sets. The two best configurations both used Number of Segments (Num Seg) = 16 and Window Size = 4. Subsequent analysis focuses on verifying the effectiveness of these values and determining the optimal Area Size.

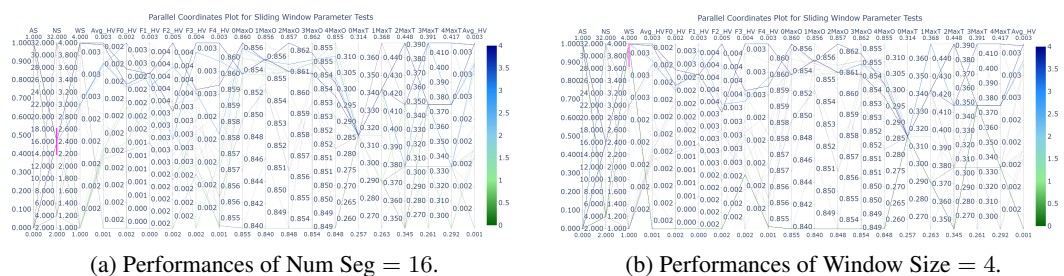

(a) Performances of Num Seg = 16.

(b) Performances of Window Size = 4.

Figure 18: Sliding Window Num Seg and Window Size Parameter Experiments.

Figure 18a shows that Num Seg = 16 yields both top and suboptimal results, with consistently good performance when Window Size ¿ 1. Similarly, Figure 18b demonstrates that Window Size = 4 produces good results when Num Seg ≠ 4. These findings confirm that Num Seg = 16 and Window Size = 4 are optimal values among those tested.

Figures 19a and 19b compare the two Area Size values tested:

- FullSize: using the minimum and maximum values from the entire dataset for each dimension

(a) Performances of Area Size = FullSize.      (b) Performances of Area Size = TopicMinMax.

Figure 19: Sliding Window Area Size Parameter Experiments.

- TopicMinMax: using the minimum and maximum values only from the selected topic/class

The results show no clear advantage for either option. Therefore, we selected the Area Size that produced the higher hypervolume given Num Seg $= 16$ and Window Size $= 4$. In conclusion, the experimentally determined optimal parameter set for the Sliding Window strategy is:

- Area Size = TopicMinMax
- Num Seg = 16
- Window Size = 4

### B.2.2   HIERARCHICAL SLIDING WINDOW (HSW) PARAMETERS EXPERIMENTS

We investigated the parameters of the Hierarchical Sliding Window (HSW) strategy. Figure 20a presents an overview of the parameter experiments. The parameters under investigation are:

- Area Size (AS)
- Window Size (WS)
- Level 0 Num Seg (NS0)
- Level 1 Num Seg (NS1)
- Level 2 Num Seg (NS2, using a 3-level structure for HSW)

These parameters are represented by the five leftmost coordinates in the plot. We used the Hypervolume Indicator as the primary comparison metric, supplemented by the maximum values of both objectives in each of the 5 cross-validation folds.

Figure 20b highlights the top two parameter sets, both using Area Size = TopicMinMax and Window Size = Half of each level's Num Seg. Subsequent analysis focuses on verifying these parameter choices and determining optimal Num Seg values for each level.

Figures 21a and 21b compare FullSize and TopicMinMax Area Size values. While both show variable performance, TopicMinMax yields more top results, leading to its selection.

Figures 22a and 22b compare the Window Size of Half and 1. Window Size = Half clearly outperforms, leading to its selection.

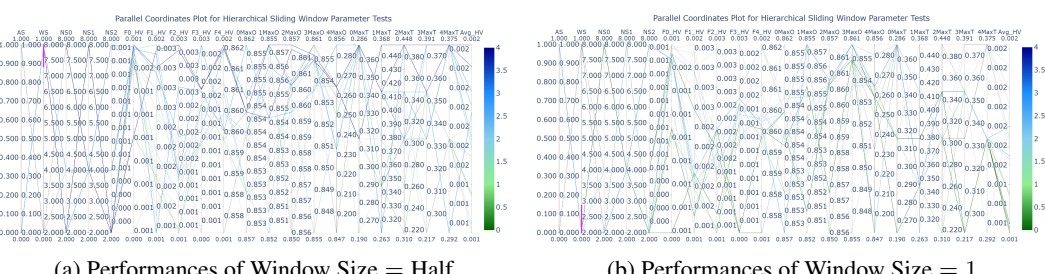

(a) Performances of Hierarchical Sliding Window Parameter Experiments.

(b) Hierarchical Sliding Window Top Results.

Figure 20: Hierarchical Sliding Window Parameter Experiments Overview and Top Results.

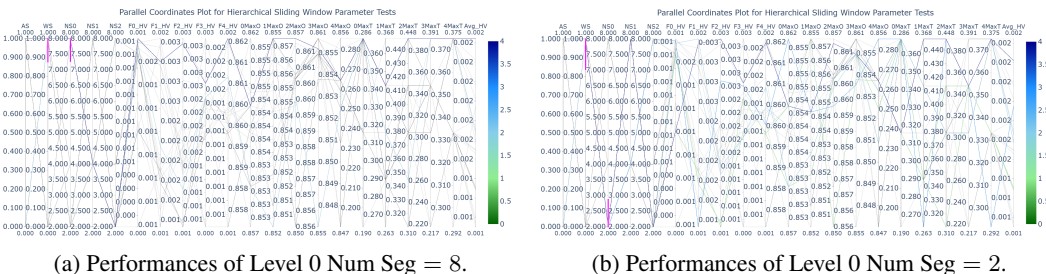

(a) Performances of Area Size = FullSize.

(b) Performances of Area Size = TopicMinMax.

Figure 21: Hierarchical Sliding Window Area Size Parameter Experiments.

(a) Performances of Window Size = Half.

(b) Performances of Window Size = 1.

Figure 22: Hierarchical Sliding Window Window Size Parameter Experiments.

(a) Performances of Level 0 Num Seg = 8.

(b) Performances of Level 0 Num Seg = 2.

Figure 23: Hierarchical Sliding Window Level 0 Number of Segments Parameter Experiments.

Figures 23a and 23b compare Level 0 Num Seg values of 8 and 2, when Window Size = Half. Num Seg = 8 shows more consistent good performance, leading to its selection.

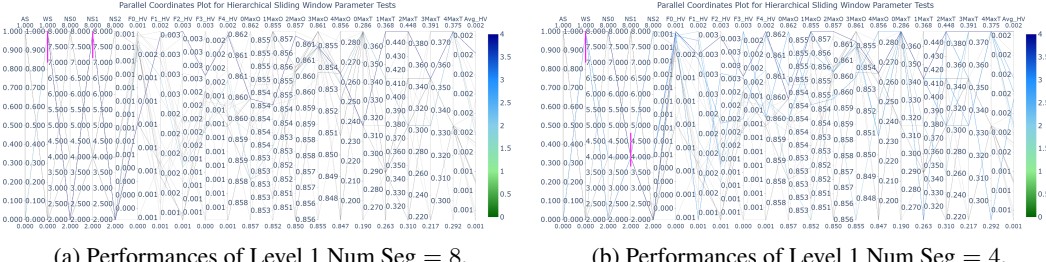

(a) Performances of Level 1 Num Seg = 8.   (b) Performances of Level 1 Num Seg = 4.

Figure 24: Hierarchical Sliding Window Level 1 Number of Segments Parameter Experiments.

Figures 24a and 24b compare Level 1 Num Seg values of 8 and 4. While both yield top results, Num Seg = 4 shows better overall performance, leading to its selection.

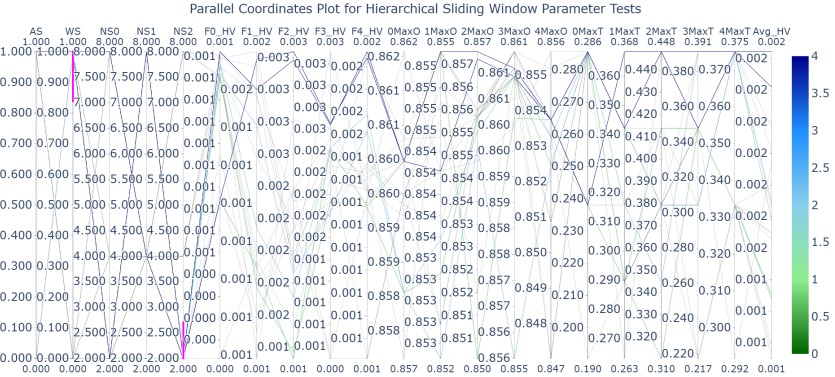

Figure 25: Performances of Level 2 Num Seg = 2.

Figure 25 shows results for Level 2 Num Seg = 2. Performance is consistently good when Level 1 Num Seg $\neq$ 2, aligns with our previous parameter choices. Therefore, with Level 0 Num Seg = 8 and Level 1 Num Seg = 4, Level 2 Num Seg = 2 will provide top results.

In conclusion, the optimal parameter set for the Hierarchical Sliding Window strategy is:

- Area Size = TopicMinMax
- Window Size = Half of each Level's Num Seg
- Level 0 Num Seg = 8
- Level 1 Num Seg = 4
- Level 2 Num Seg = 2

### B.2.3    GENETIC ALGORITHM (GA) PARAMETERS EXPERIMENTS

We investigated the parameters for the Genetic Algorithm (GA) strategy. Figure 26 presents an overview of the parameter experiments. The parameters under investigation are:

- Population Size and Selection Size (PSSize)

- Crossover Rate and Initial Mutation Rate (CMRate)

These parameters are represented by the four leftmost coordinates in the plot. We used the Hypervolume Indicator as the primary comparison metric, supplemented by the maximum values of both objectives in each of the 5 cross-validation folds.

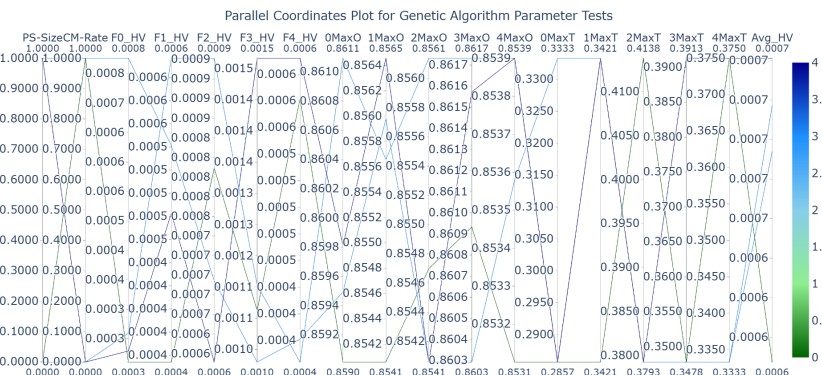

Figure 26: Genetic Algorithm Parameters Experiment Overview (PSSize: Population-Selection Size, CMRate: Crossover-Mutation Rate).

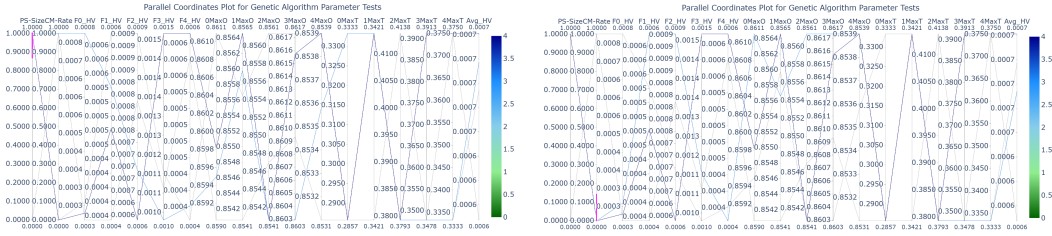

(a) Performances of Population and Selection Sizes of 20 and 20.

(b) Performances of Crossover and Initial Mutation rates of 0.7 and 0.3.

Figure 27: Genetic Algorithm Parameters Experiments.

Figure 27a illustrates the performance when both population size and selection size are set to 20. These results consistently outperform the alternative configuration of population size 20 and selection size 10.

Figure 27b shows the performance with a crossover rate of 0.7 and an initial mutation rate of 0.3. This configuration demonstrates superior performance compared to the alternative rates of 0.9 and 0.1, respectively, when population and selection sizes are held constant.

Based on these experiments, we determined the optimal parameter set for the Genetic Algorithm strategy:

- Population Size = 20

- Selection Size = 20

- Crossover Rate = 0.7

- Initial Mutation Rate = 0.3

## C ALGORITHMS

### C.1 EXAMPLES SUBSET SELECTION STRATEGIES

Based on the parameter experiments reported in Appendix B, we finalized the workflows for the three example selection algorithms. To complement the description provided in Section 3.3, we present here the detailed workflows for the Hierarchical Sliding Window (HSW) and Genetic Algorithm (GA) strategies.

---

**Algorithm 1** Hierarchical Sliding Window Selection Strategy

---

**Require:** $a\_s$ – x and y range of each PCA plot; $n\_s$ – list of number of segments for each level; $w\_s$ – list of window sizes for each level; $data\_syn$ – synthetic data; $class$ – the selected class; $l$ – level of hierarchical sliding window

**Ensure:** Best windows found on each level

1: **for** each $plot$ in $PCA\_plots$ **do**
2:    Initialize $l \leftarrow 0$
3:    Initialize $selected\_windows \leftarrow \{a\_s\}$
4:    **while** $l <$ length of $n\_s$ **and** not terminated **do**
5:      $best\_windows \leftarrow \{\}$
6:      **for** each $area$ in $selected\_windows$ **do**
7:        Perform sliding window on $area$ using $n\_s[l]$ and $w\_s[l]$
8:        **for** each $window$ in sliding window **do**
9:          Retrieve data dots from $window$ belonging to $class$
10:          $syn\_samples \leftarrow$ AutoGeTS(data dots, $data\_syn$)
11:          Train classification model using $syn\_samples$
12:          Evaluate model and compute performance metric $J'(W)$
13:          Record performance metric as the score for $window$
14:        **end for**
15:        Add $window$ with maximum score to $best\_windows$
16:      **end for**
17:      $selected\_windows \leftarrow best\_windows$
18:      $l \leftarrow l + 1$
19:      **if** termination condition met **then**
20:        Set terminated to True
21:      **end if**
22:    **end while**
23: **end for**
24: **return** $selected\_windows$

---

Algorithm 1 outlines the Hierarchical Sliding Window (HSW) selection strategy. This algorithm iteratively refines the search space across multiple levels, efficiently identifying optimal windows for synthetic data generation.

Algorithm 2 details the Genetic Algorithm (GA) selection strategy. This evolutionary approach uses fitness-based selection, crossover, and mutation operations to optimize the set of data examples used for synthetic data generation.

### C.2 ENSEMBLE MULTI-CLASS ALGORITHMS

This section presents detailed algorithms for the ensemble strategies outlined in Section 3.4. These algorithms are designed to address specific business requirements identified in Section 3.2.

Algorithm 3 addresses Requirement 1, focusing on improving the performance of underperforming classes. Algorithm 4 targets Requirement 2, aiming to enhance

**Algorithm 2** GA Selection-Generation-Retraining Approach

---

**Require:** $n$ – population size; $g_{max}$ – maximum generations; $F(S)$ – fitness score from objective function; *mutation* – representation mutation function; *crossover* – representation crossover function; $p_{cro}$ – crossover probability; $p_{mut}$ – mutation probability.

**Ensure:** The final individual(s) maximises the fitness score

1: $P_0 \leftarrow$ randomly generated population of selected data examples of size $n$ with priority value based chromosome representation

2: $F_0 \leftarrow \{F(P_0[i]) \mid i \in 1, \ldots, n\}$ {Evaluate initial population fitness through AutoGeTS process}

3: $G \leftarrow 0$ {Generation counter}

4: **while** $G < g_{max}$ **do**

5:     Select individuals for the mating pool using Lexicase and Clustered tournament selection based on fitness score

6:     $P' \leftarrow$ Generate offspring using weight mapping *crossover* and adaptive polynomial *mutation* with probability $p_{cro}$ and $p_{mut}$

7:     Evaluate offspring: $F' \leftarrow \{F(P'[i]) \mid i \in 1, \ldots, n\}$ {Evaluate new population}

8:     Combine parent and offspring populations: $R \leftarrow P_G \cup P'$

9:     Select the next generation $P_{G+1}$ from $R$ using elitism.

10:     $G \leftarrow G + 1$ {Increment generation counter}

11: **end while**

12: Return the best set(s) of data examples in $P_G$ based on fitness score

---

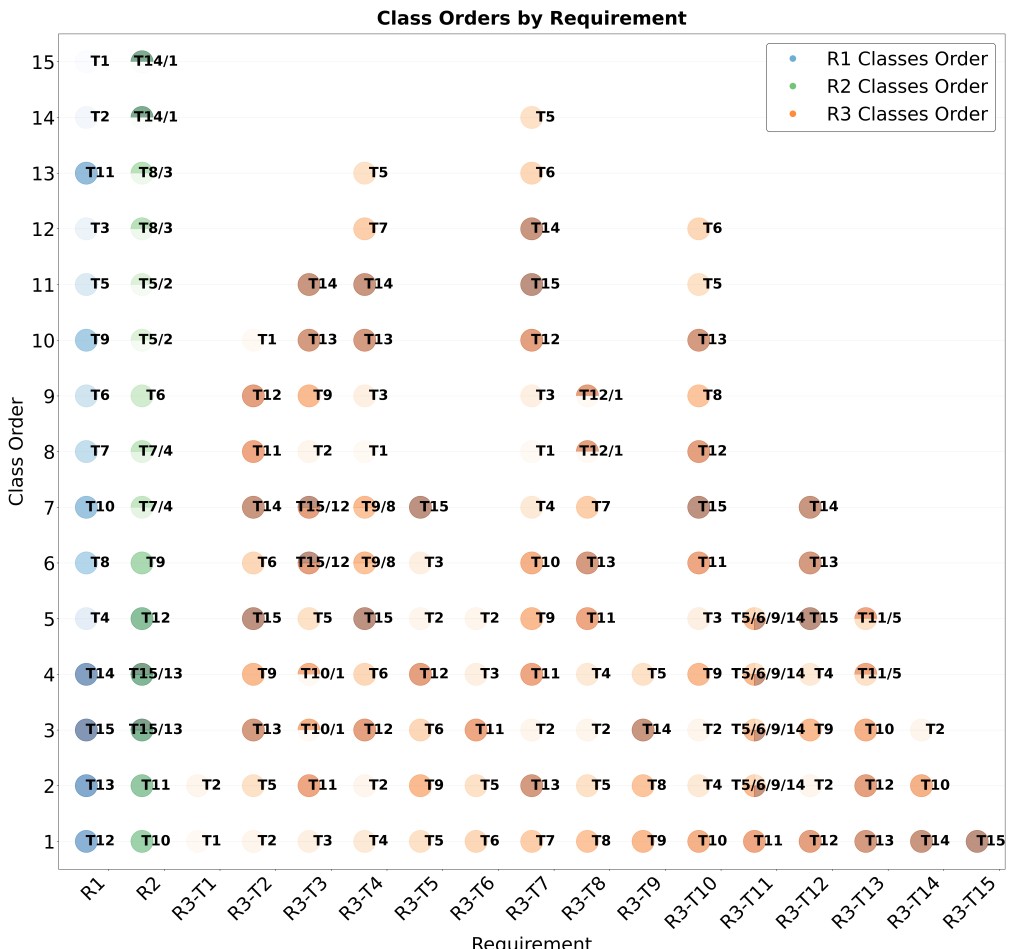

Figure 28: Class Orders for R1 (improve each class), R2 (improve overall performance), and R3 (improve an important class).

overall classification performance. Algorithm 5 addresses Requirement 3, which prioritizes improving the performance of a specific important class.

Each algorithm iteratively utilizes the AutoGeTS process, incorporating insights from our experimental results, including the strategy-objective combinations from Table 3 and class relationships and class order from Figures 6b and 6a. In Figure 6a, each row represents the application of AutoGeTS to one of the fifteen classes, while each column shows the corresponding improvement in class balanced accuracy for that particular class. The resulting class orderings for the three requirements are illustrated in Figure 28. The blue column depicts the class ordering for Requirement 1 (improving all individual class performance), where the bottom-most dot represents the first-ordered class and the top-most dot indicates the last-ordered class. The green column shows the class ordering for Requirement 2 (improving overall performance), while the orange columns display the class ordering for Requirement 3 (improving a specific important class), with each R3 column corresponding to a designated important class.

---

**Algorithm 3** Requirement 1: Improve Bad Performing Classes

---

**Require:** Classes with performance metrics, AutoGeTS process, GA parameters
**Ensure:** Improved classification model for bad-performing classes
 1:  Sort classes by class size in ascending order
 2:  **for** Iteration $i$, each class $C_i$ with class balanced accuracy $< 0.8$ **do**
 3:      Select strategy-objective that best improves $C_i$ balanced accuracy from Lookup Table 3
 4:      Apply AutoGeTS to $C_i$
 5:      **if** improvement achieved **then**
 6:          Record maximum improvement and corresponding model $m$ for $C_i$
 7:          **if** exist $M$ maintain improvement in each $C_{n<i}$ balanced accuracy by at least 50% **then**
 8:              Select model $m$ from $M$ that best improves $C_i$'s balanced accuracy
 9:          **else**
10:              Terminate algorithm
11:          **end if**
12:          Append the synthetic sample from $m$ to the training set
13:      **else**
14:          Terminate algorithm
15:      **end if**
16:  **end for**

---

**Algorithm 4** Requirement 2: Improve Overall Performance

---

**Require:** Classes with performance metrics, AutoGeTS process, GA parameters
**Ensure:** Improved overall classification performance
 1:  **while** overall performance can be improved **do**
 2:      Select a class $C$ in descending order of improving overall performance based on figure 6b.
 3:      Select strategy-objective combination that best improves global metrics for $C$ from Lookup Table 3
 4:      Apply AutoGeTS to $C$
 5:      Record maximum improvement and corresponding model $m$ for $C$
 6:      Append the synthetic sample from $m$ to the training set
 7:      **if** overall performance not improved **then**
 8:          Terminate algorithm
 9:      **end if**
10:  **end while**

---

---

**Algorithm 5** Requirement 3: Improve Important Class

---

**Require:** Classes with performances, AutoGeTS process, GA parameters, Important class $IC$
**Ensure:** Improved important class performance
 1: Identify related classes $RC$ of $IC$ from figure 6a.
 2: Sort $IC$ and $RC$ in descending order of $IC$ improvement according to figure 6a.
 3: **for** iteration $i$, each class $IC_i$ or $RC$ in the order **do**
 4:    Apply AutoGeTS to $IC_i$ or $RC$ with the strategy-objective combination that best improves its local metrics according to Lookup Table 3
 5:    Record maximum improvement in $IC_i$'s class performance and corresponding model $m$
 6:    Append the synthetic sample from $m$ to the training set
 7:    **if** $IC$ performance not improved **then**
 8:       Terminate algorithm
 9:    **end if**
10: **end for**

---

## D  FIXED-TIME EXPERIMENTS

### D.1  FIXED-TIME EXPERIMENTS: IMPROVING LOCAL VS GLOBAL METRIC

Following our analysis of the Performance Improvement Overview in Section 4.2 and before comparing strategies and objectives in Section 4.3, we investigated whether AutoGeTS could simultaneously improve both class-specific and overall performance, and to what extent these goals might be contradictory. To this end, we compared models trained with synthetic data that achieved maximum improvements in either the local metric (Class Balanced Accuracy) or the global metric (Overall Balanced Accuracy) for all 15 classes.

Analysis of Figure 29 reveals significant insights into AutoGeTS's performance. When optimizing for local metrics, 11 out of 15 classes (excluding T1, T7, T8, and T14) showed improvements without negatively impacting the global metric. Only T7 and T8 exhibited relatively larger decreases ($-0.1\%$) in global performance when local performance was maximized. Similarly, when optimizing for global metrics, 11 out of 15 classes (excluding T1, T9, T11, and T14) demonstrated improvements without compromising local performance. In this case, only T9 showed a relatively larger decrease (-1%) in local performance when global performance was maximized. These observations demonstrate AutoGeTS's capability to simultaneously improve both local and global metrics in the majority of cases, confirming its effectiveness in addressing both class-specific and overall performance goals.

However, the instances where trade-offs occurred between local and global performance suggest the need for future research into advanced optimization methods. Such research could explore both example selection strategies and objective functions capable of optimizing multiple objectives simultaneously, potentially eliminating these trade-offs and further enhancing AutoGeTS's performance across all classes.

### D.2  FIXED-TIME EXPERIMENTS: PERFORMANCE TRAJECTORIES OVER RETRAINING TIME

Following the comparison of the three example selection strategies in Section 4.3, we further analyzed their performance improvements with respect to retraining

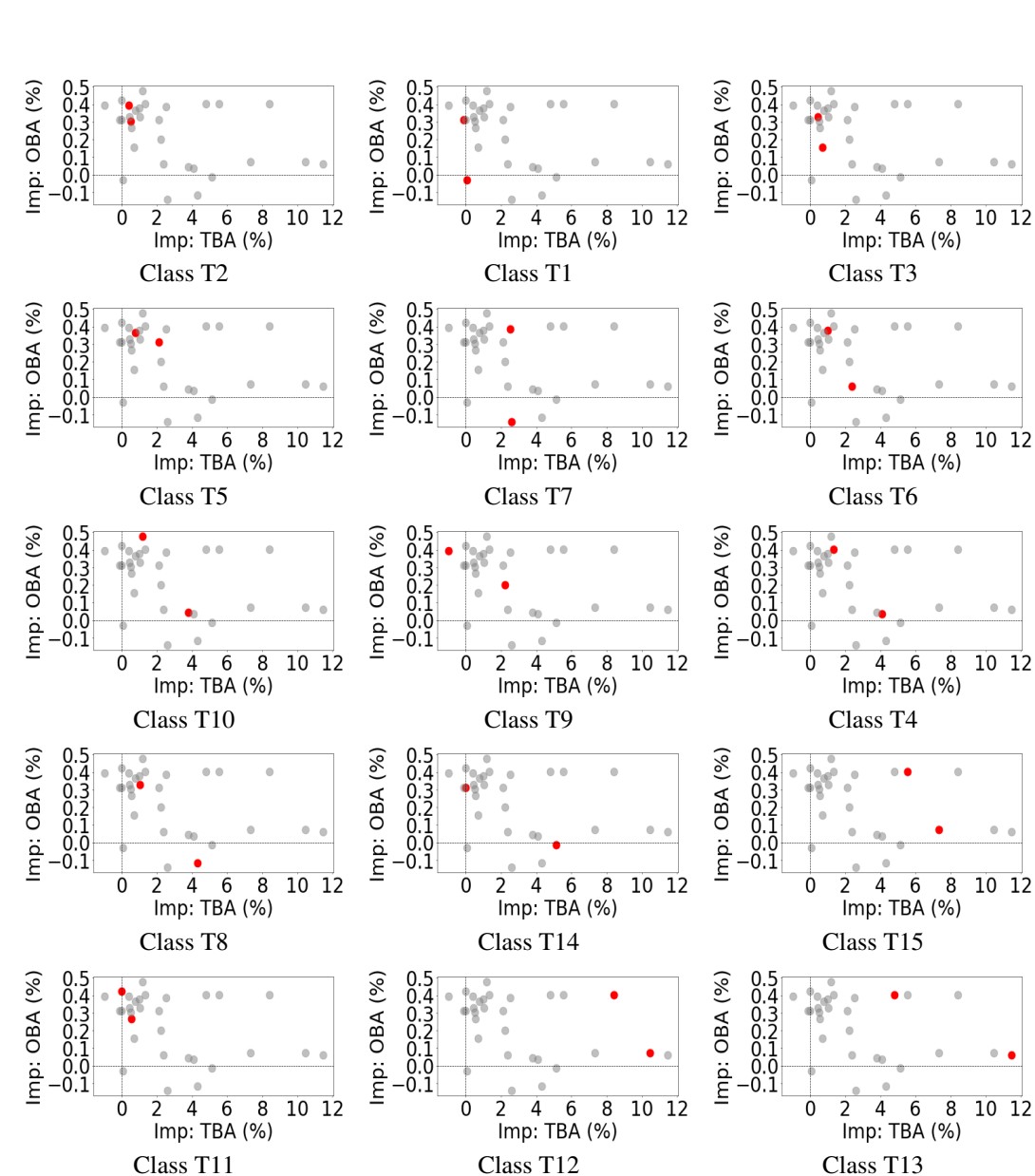

Figure 29: Models Found with Maximum Improvements in Class Balanced Accuracy or Overall Balanced Accuracy for each of 15 Classes.

time. Four experiments were conducted, each maximizing a different metric: Class Recall, Class Balanced Accuracy, Overall Balanced Accuracy, and Overall F1-Score. The three example search strategies and 15 classes served as independent variables, with a constraint of 1 hour total GPU running time.

Figure 30 compares improvements in Class Recall when maximizing Class Recall is the optimization objective. Figure 31 compares improvements in Class Balanced Accuracy when maximizing Class Balanced Accuracy is the optimization objective. Figure 32 compares improvements in Overall Balanced Accuracy when maximizing Overall Balanced Accuracy is the optimization objective. Figure 33 compares improvements in Overall F1-Score when maximizing Overall F1-Score is the optimization objective.

For these class-specific metrics, we observed that HSW often achieves its best or near-best improvements within the first 1/3 of training time, especially for classes where it performs best. GA typically reaches its peak performance within the first 1/3 of retraining time, except for T14 and T15. However, for these classes, GA outperforms other strategies even before reaching its best performance. These observations confirm the computational efficiency of HSW and GA strategies.

SW also often achieves its best performance around 1/3 of retraining time. But for classes where it outperforms other strategies (e.g., T5 and T10), SW only surpasses others quite late, starting from around 2/3 or even 4/5 of retraining time. This reflects both its advantage in avoiding plateaus and its computational inefficiency due to its brute-force nature.

Figure 33 compares improvements in Overall F1-Score when maximizing Overall F1-Score is the optimization objective.

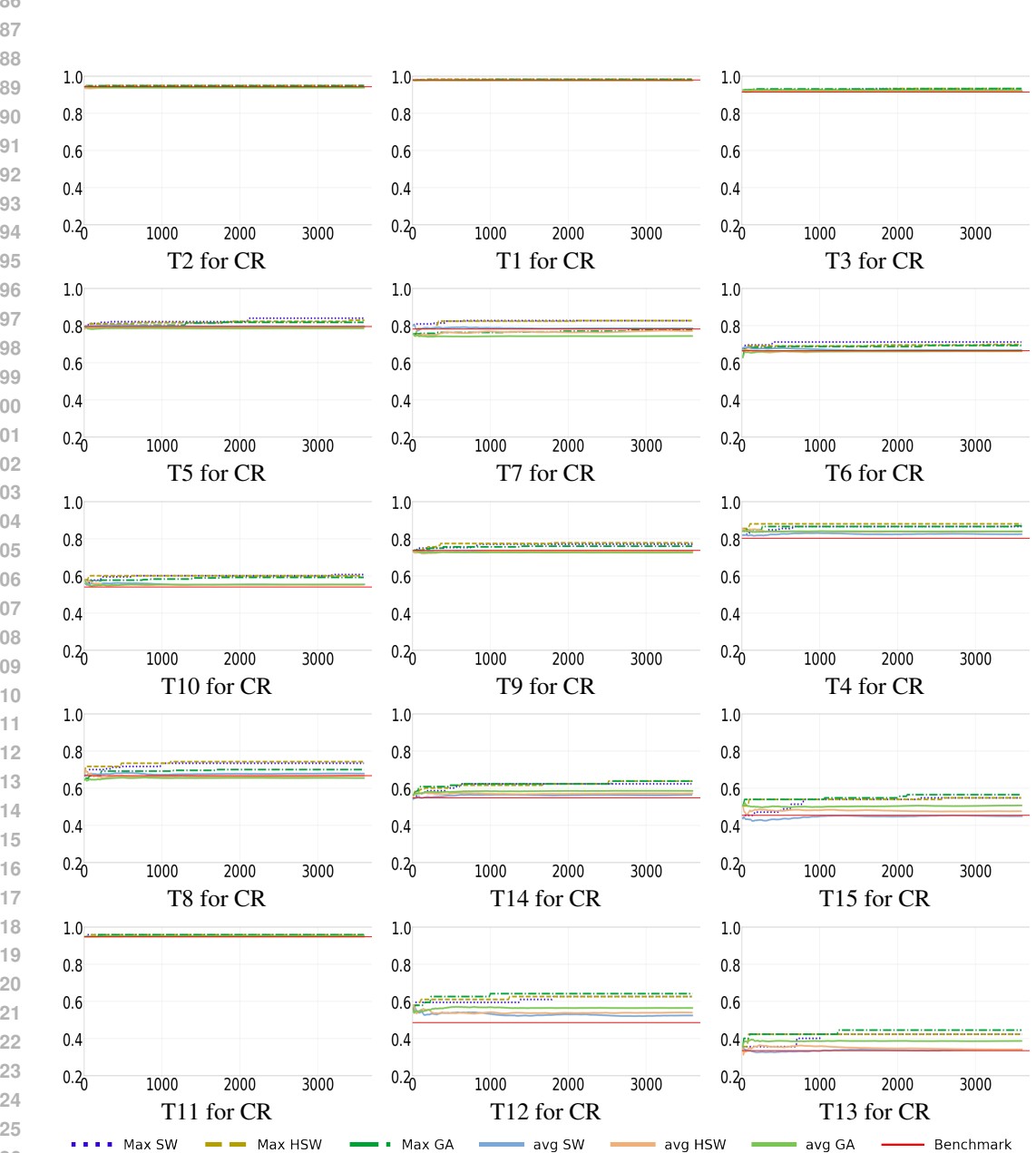

Figure 30: Fixed Time, Class Recall as Objective, Comparing on Class Recall Improvement: each time series plot has six lines. SW max, HSW max, GA max, SW avg, HSW avg, GA avg.

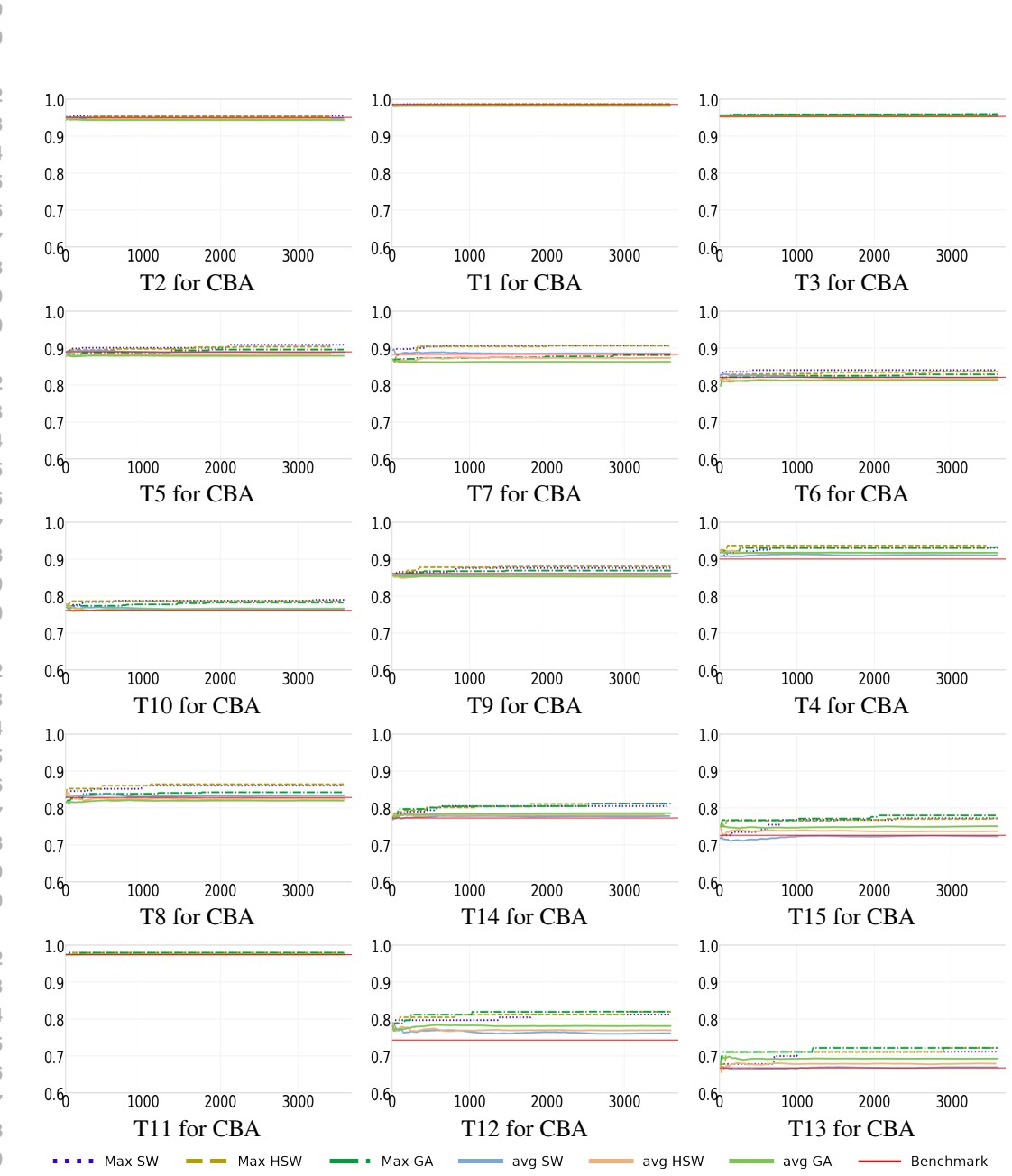

Figure 31: Fixed Time, Class Balanced Accuracy as Objective, Comparing on Class Balanced Accuracy Improvement: each time series plot has six lines. SW max, HSW max, GA max, SW avg, HSW avg, GA avg.

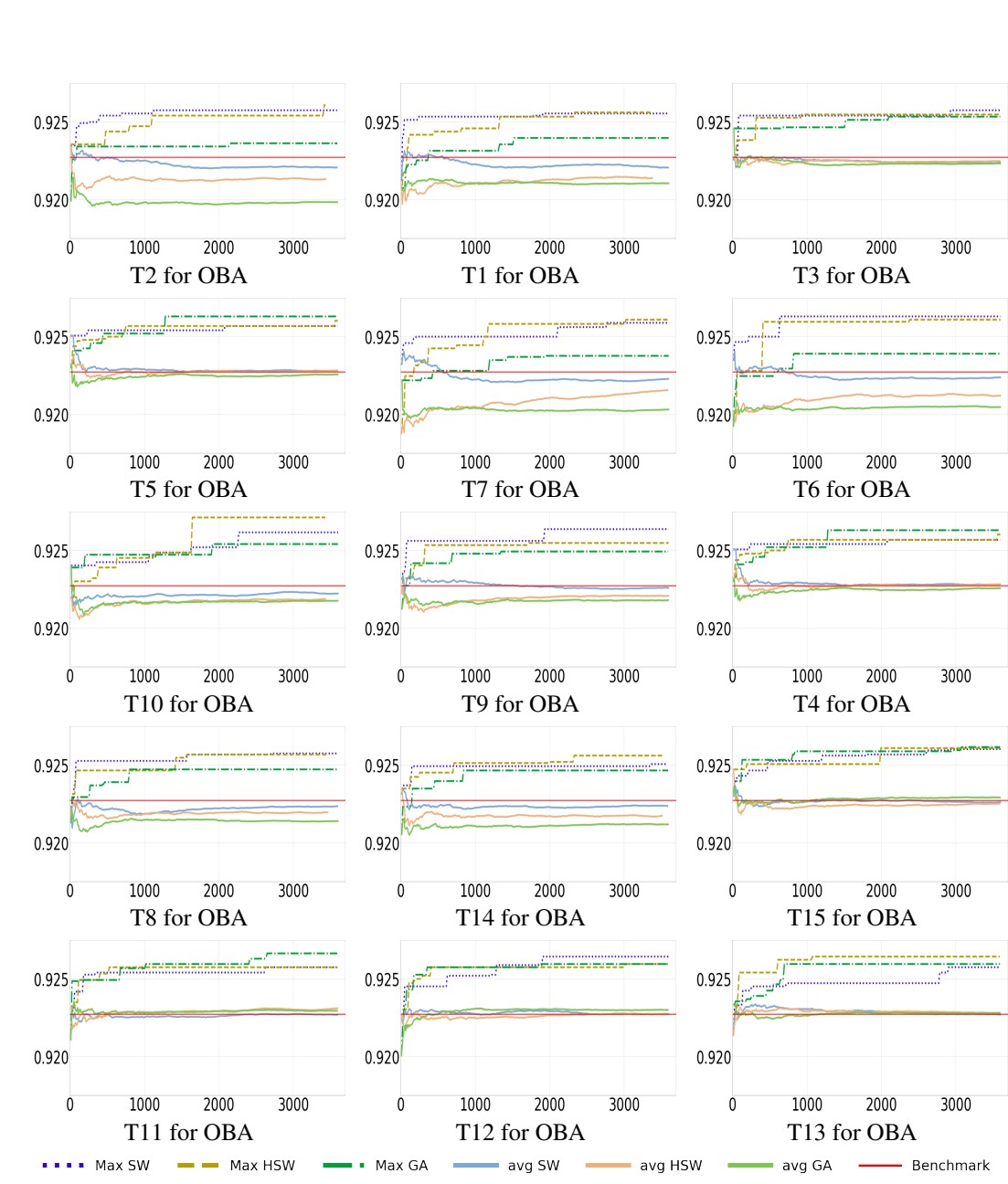

Figure 32: Fixed Time, Overall Balanced Accuracy as Objective, Comparing on Overall Balanced Accuracy Improvement: each time series plot has six lines. SW max, HSW max, GA max, SW avg, HSW avg, GA avg.

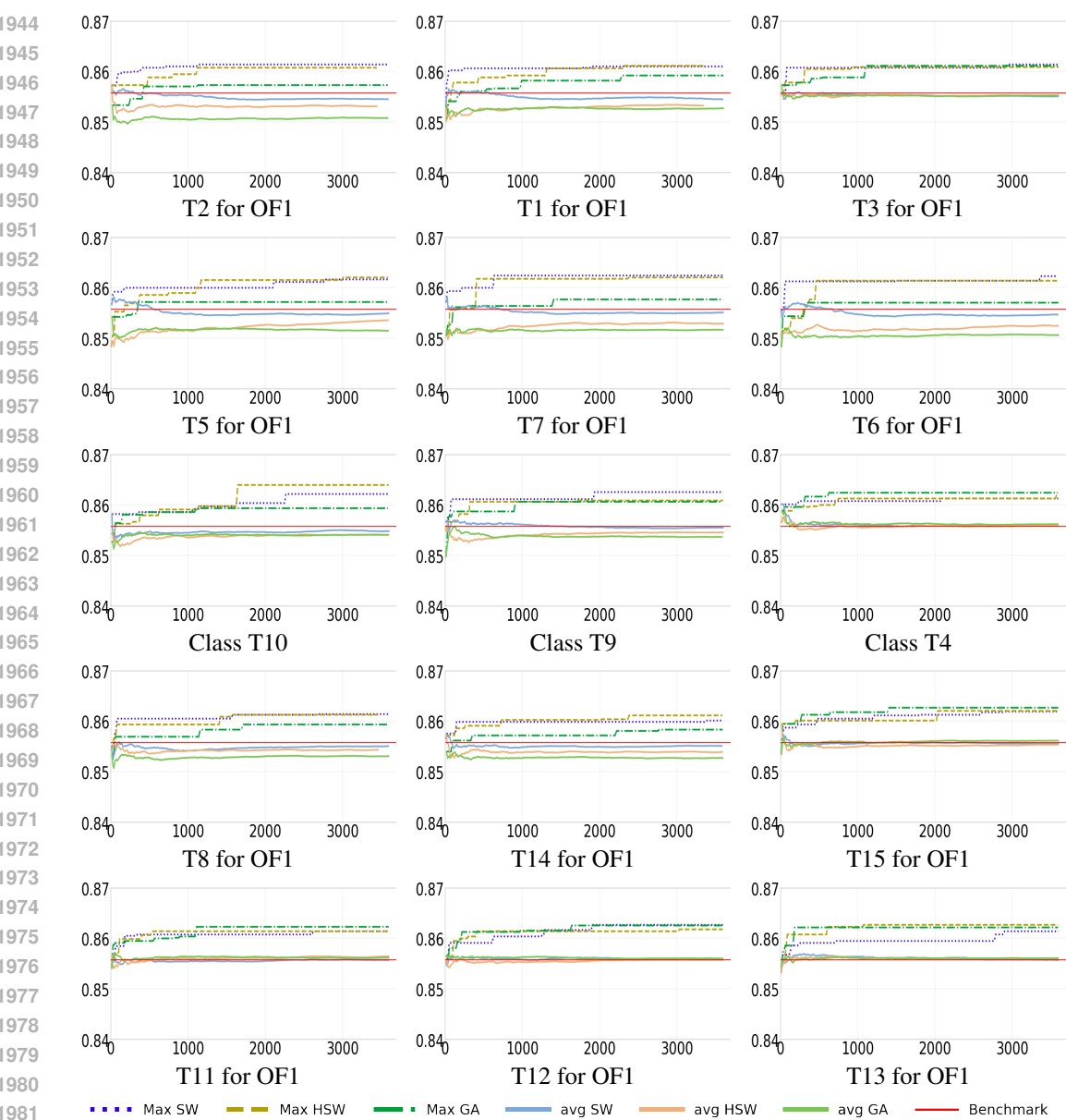

Figure 33: Fixed Time, Overall F1-Score as Objective, Comparing on Overall F1-Score Improvement: each time series plot has six lines. SW max, HSW max, GA max, SW avg, HSW avg, GA avg.

For these global metrics, we observed that all three strategies often required more than 1/3 of retraining time to reach their best or near-best performance, especially for Overall Balanced Accuracy. This suggests improving global metrics would be more computationally complex than improving local metrics for all three strategies, as it would require a more synergistic effect.

When improving global metrics, GA often leads in performance gain from early on (around 1/3 retraining time) for classes where it obtains the largest improvement. SW and HSW show more variable timing in achieving leading performance, ranging from early to late in the retraining process. GA's performance pattern con-

firms its evolutionary nature, demonstrating an ability to maintain and exploit good solutions once found.

## E BENCHMARK DATA AUGMENTATION COMPARISON EXPERIMENTS

To provide an additional baseline for evaluating the AutoGeTS workflow, we implemented traditional data augmentation methods, alongside the LLM-based approach, as the generator in the synthetic data generation process. Specifically, we utilized the Easy Data Augmentation (EDA) tool (Wei & Zou (2019)), which incorporates techniques such as synonym replacement, random insertion, random swap, and random deletion. The evaluation followed our established experimental protocol of 1 GPU hour fixed-time experiments to improve M0, employing Random, SW, HSW, and GA selection strategies, each executed separately with either maximizing Overall Balanced Accuracy (OBA) or Class Balanced Accuracy (CBA) as the optimization objective. All experimental parameters remained consistent with those described in Section 4.

### E.1 BEST PERFORMANCE COMPARISON: OBA & CBA IMPROVEMENTS

Table 7: Overall Balanced Accuracy (Global) Comparison between EDA and GPT-3.5 as Generator.

| Class Name | Random | | Sliding Window | | Hierarchical SW | | Genetic Algorithm | |
|---|---|---|---|---|---|---|---|---|
| | EDA | GPT-3.5 | EDA | GPT-3.5 | EDA | GPT-3.5 | EDA | GPT-3.5 |
| T2 | ▲0.0001 | ▲0.0012 | ▲0.0012 | ▲0.0030 | ▲0.0014 | ▲0.0034 | ▲0.0004 | ▲0.0009 |
| T1 | ▲0.0006 | ▲0.0009 | ▲0.0017 | ▲0.0028 | ▲0.0024 | ▲0.0029 | ▲0.0012 | ▲0.0018 |
| T3 | ▲0.0006 | ▲0.0014 | ▲0.0011 | ▲0.0030 | ▲0.0018 | ▲0.0027 | ▲0.0010 | ▲0.0029 |
| T5 | ▲0.0003 | ▲0.0002 | ▲0.0014 | ▲0.0032 | ▲0.0011 | ▲0.0034 | ▲0.0006 | ▲0.0010 |
| T7 | ▲0.0006 | ▲0.0018 | ▲0.0009 | ▲0.0036 | ▲0.0015 | ▲0.0034 | ▲0.0006 | ▲0.0012 |
| T6 | ▲0.0010 | ▲0.0016 | ▲0.0010 | ▲0.0035 | ▲0.0013 | ▲0.0030 | ▲0.0013 | ▲0.0015 |
| T10 | ▼0.0011 | ▼0.0006 | ▲0.0008 | ▲0.0034 | ▲0.0015 | ▲0.0044 | ▲0.0016 | ▲0.0027 |
| T9 | ▲0.0001 | ▲0.0016 | ▲0.0019 | ▲0.0036 | ▲0.0014 | ▲0.0027 | ▲0.0011 | ▲0.0026 |
| T4 | ▲0.0010 | ▲0.0018 | ▲0.0005 | ▲0.0029 | ▲0.0019 | ▲0.0033 | ▲0.0018 | ▲0.0036 |
| T8 | ▲0.0006 | ▲0.0012 | ▲0.0008 | ▲0.0030 | ▲0.0011 | ▲0.0029 | ▲0.0014 | ▲0.0020 |
| T14 | ▲0.0006 | ▲0.0009 | ▲0.0014 | ▲0.0023 | ▲0.0019 | ▲0.0029 | ▲0.0003 | ▲0.0019 |
| T15 | ▲0.0001 | ▲0.0005 | ▲0.0011 | ▲0.0033 | ▲0.0013 | ▲0.0034 | ▲0.0022 | ▲0.0037 |
| T11 | ▲0.0012 | ▲0.0020 | ▲0.0014 | ▲0.0030 | ▲0.0014 | ▲0.0030 | ▲0.0022 | ▲0.0039 |
| T12 | ▲0.0010 | ▲0.0014 | ▲0.0011 | ▲0.0037 | ▲0.0021 | ▲0.0032 | ▲0.0020 | ▲0.0036 |
| T13 | ▲0.0004 | ▲0.0006 | ▲0.0013 | ▲0.0030 | ▲0.0011 | ▲0.0037 | ▲0.0022 | ▲0.0034 |

Table 8: Class Balanced Accuracy (Local) Comparison between EDA and GPT-3.5 as Generator.

| Class Name | Random | | Sliding Window | | Hierarchical SW | | Genetic Algorithm | |
|---|---|---|---|---|---|---|---|---|
| | EDA | GPT-3.5 | EDA | GPT-3.5 | EDA | GPT-3.5 | EDA | GPT-3.5 |
| T2 | ▼0.0212 | ▼0.0155 | ▲0.0003 | ▲0.0050 | ▲0.0003 | ▲0.0048 | ▼0.0011 | ▼0.0010 |
| T1 | ▼0.0219 | ▼0.0202 | ▼0.0030 | ▲0.0005 | ▼0.0027 | ▲0.0005 | ▼0.0028 | ▼0.0018 |
| T3 | ▼0.0215 | ▼0.0148 | ▲0.0015 | ▲0.0058 | ▲0.0007 | ▲0.0062 | ▼0.0012 | ▲0.0069 |
| T5 | ▲0.0001 | ▼0.0136 | ▲0.0139 | ▲0.0189 | ▲0.0149 | ▲0.0140 | ▲0.0048 | ▲0.0059 |
| T7 | ▼0.0082 | ▼0.0091 | ▲0.0191 | ▲0.0228 | ▲0.0180 | ▲0.0226 | ▲0.0056 | ▼0.0026 |
| T6 | ▼0.0055 | ▼0.0124 | ▲0.0122 | ▲0.0190 | ▲0.0195 | ▲0.0196 | ▲0.0093 | ▲0.0073 |
| T10 | ▲0.0013 | ▲0.0051 | ▲0.0244 | ▲0.0281 | ▲0.0255 | ▲0.0247 | ▲0.0203 | ▲0.0208 |
| T9 | ▼0.0174 | ▲0.0049 | ▲0.0041 | ▲0.0147 | ▲0.0071 | ▲0.0191 | ▼0.0023 | ▲0.0077 |
| T4 | ▲0.0138 | ▲0.0118 | ▲0.0293 | ▲0.0304 | ▲0.0272 | ▲0.0369 | ▲0.0340 | ▲0.0323 |
| T8 | ▼0.0000 | ▲0.0146 | ▲0.0181 | ▲0.0321 | ▲0.0285 | ▲0.0358 | ▲0.0242 | ▲0.0142 |
| T14 | ▼0.0014 | ▲0.0029 | ▲0.0153 | ▲0.0326 | ▲0.0221 | ▲0.0395 | ▲0.0185 | ▲0.0396 |
| T15 | ▲0.0038 | ▲0.0067 | ▲0.0245 | ▲0.0456 | ▲0.0409 | ▲0.0446 | ▲0.0199 | ▲0.0533 |
| T11 | ▼0.0143 | ▼0.0248 | ▲0.0051 | ▲0.0054 | ▲0.0052 | ▲0.0053 | ▲0.0051 | ▲0.0053 |
| T12 | ▲0.0405 | ▲0.0472 | ▲0.0656 | ▲0.0699 | ▲0.0705 | ▲0.0772 | ▲0.0709 | ▲0.0775 |
| T13 | ▲0.0204 | ▲0.0237 | ▲0.0415 | ▲0.0443 | ▲0.0514 | ▲0.0548 | ▲0.0392 | ▲0.0548 |

As our deployment strategy selects the best-performing model for each target class based on specified objectives, we analyze the maximum OBA and CBA improvements achieved and compare them with the AutoGeTS performance presented in Section 4.2. Tables 7 and 8 present these comparative results.

The comparison between EDA and LLM (GPT-3.5) approaches reveals consistent performance advantages for the LLM-based AutoGeTS workflow across all selection strategies. With the random selection, EDA's best-performing models performed below LLM's by margins of 0.06% in OBA and 0.25% in CBA, underperforming in 15 and 11 classes respectively. This performance gap widened with strategic selection methods: under SW, EDA performed 0.20% lower in OBA and 0.69% lower in CBA than LLM, with inferior results across all 15 classes for both metrics. Similarly, with HSW, EDA showed 0.17% lower OBA and 0.51% lower CBA on average, underperforming in 15 and 13 classes respectively. The GA strategy yielded comparable results, with EDA's best-performing models averaging 0.11% below LLM in OBA and 0.51% below in CBA, showing lower performance in 15 and 11 classes respectively.

### E.2 COMPARISON OF OBA IMPROVEMENTS OVER TIME

We extend our analysis to examine the temporal progression of overall balanced accuracy (OBA) improvements across all 15 classes, comparing EDA and LLM (GPT-3.5) results within the 1-GPU hour experimental constraint.

Figure 34 presents these comparisons, displaying the results of four selection strategies (SW, HSW, GA, and random selection) for both approaches. The EDA results are depicted with solid lines, while the corresponding LLM results are shown with dotted or dashed lines of the same color. Across all 15 classes, each EDA trajectory (solid line) mostly falls below its LLM counterpart (dotted or dashed line), demonstrating the superior overall performance of the LLM-based AutoGeTS workflow.

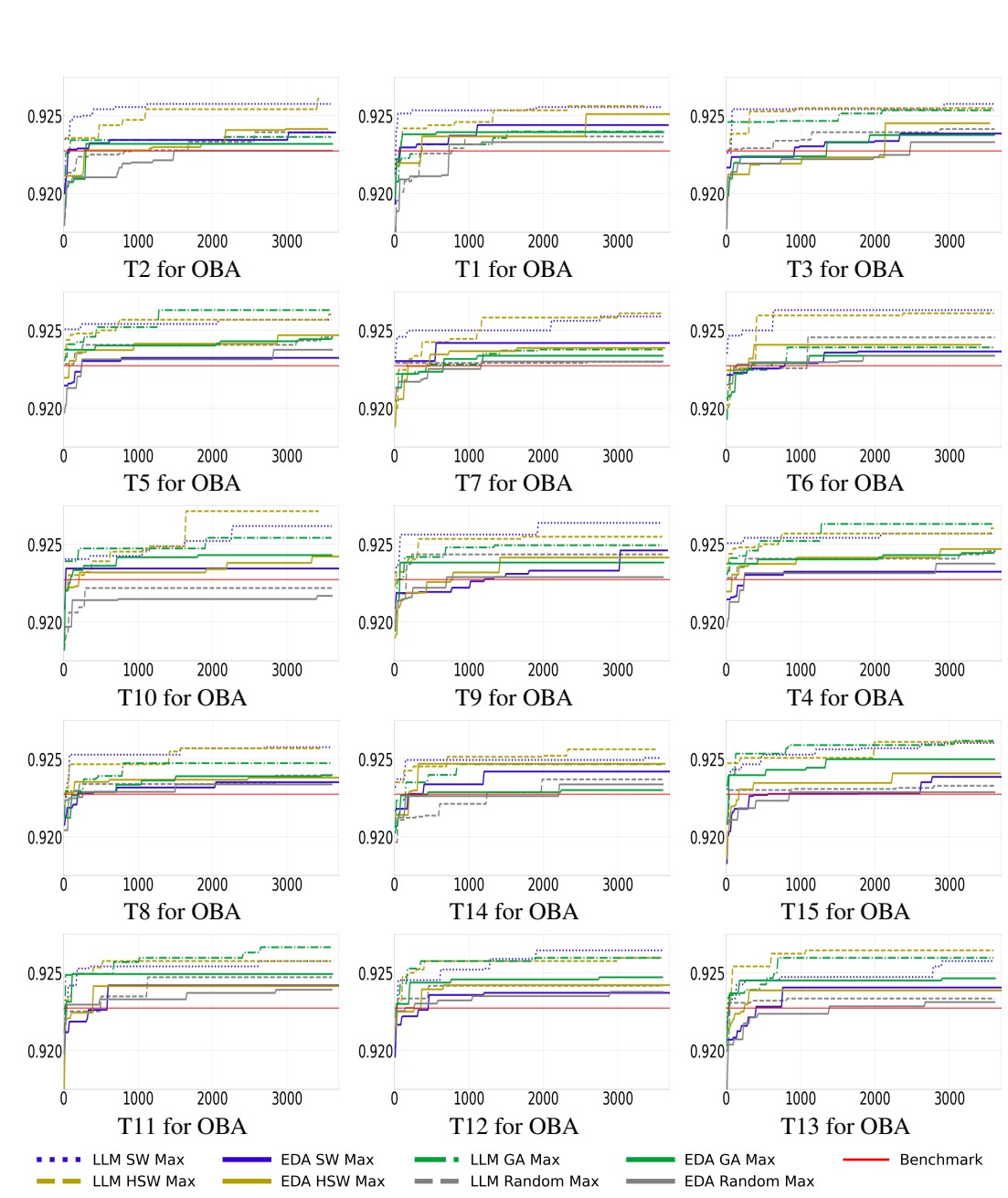

Figure 34: Fixed 1 Hour GPU time (x-axis, in seconds), Comparing on OBA Improvement (y-axis): solid lines are from EDA and dotted lines are from GPT-3.5.

