# OpenReview forum: "AutoGeTS: Automated Generation of Text Synthetics for Improving Text Classification"
_ICLR.cc/2025/Conference — Submitted to ICLR 2025_

### Official Review · Reviewer_qNuX · 2024-11-02

**Soundness:** 3
**Presentation:** 2
**Contribution:** 3
**Rating:** 5
**Confidence:** 3

**Summary:**

The paper introduces a framework for using synthetic data generated by LLMs to improve text classification tasks, particularly for low-frequency classes, without the need for human intervention. They address the challenges of limited data for underrepresented classes with the framework of automatic selection. They explored three methods for selecting inputs, which are sliding window (SW), hierarchical sliding window (HSW), and genetic algorithm (GA). The methods target specific objectives, for example, class recall, class-based balanced accuracy, overall balanced accuracy, or F1-score. Experimental results showed that HSW is effective for larger classes and GA performs well for smaller and mid-sized classes. They also proposed an ensemble algorithm that combines these methods to further improve performance.

**Strengths:**

- The paper introduces a framework that addresses the real-world challenges of class imbalance in text classification, a common issue in industrial applications, by leveraging synthetic data from LLMs to improve performance in low-frequency classes. The framework automates the selection of inputs for synthetic data generation, reducing the need for domain expertise.

- The paper tested three different strategies to gain insights into which methods work well under various conditions. Also, they considered ensemble algorithm for further improving performance.

**Weaknesses:**

- The paper evaluates their proposed framework on only a single dataset, which limits the demonstration of AutoGets's effectiveness in other domains.  Additionally, the absences of baseline comparisons, for example, traditional data augmentation techniques, makes it challenging to fully understand the advantages of AutoGeTs. While HSW and GA methods are intended to reduce computation costs, further analysis is needed to determine how computational demands scale with larger datasets, as real-world industrial applications typically involve substantial data volumes.

**Questions:**

.

---

> ### Author Response · Authors · 2024-11-20
> **This block addresses the first two weaknesses.**
>
> Thank you very much for summarizing the contributions and the strengths of the work comprehensively. The reviewer’s query is numbered by continuing these of Reviewer PqjL(1) and Reviewer dzCV(2):
>
> > **Q10 (W1): The paper evaluates their proposed framework on only a single dataset, which limits the demonstration of AutoGets's effectiveness in other domains.**
>
> **A10:** Reviewer dzCV(2) also queried this (Q7). Here we repeat our answer A7:
> We also experimented with other public datasets, such as TREC-6 (Li & Roth, 2022) and Amazon Reviews’2023 (Hou et al., 2024). We found that the improvement using synthetic data is in general feasible. However, the results as to the improvement of different classes were not comparable since the data and class labels are different. The quality of the baseline model is also a major factor. Unlike some research activities where a new model architecture, loss function, or pruning method was proposed and then tested with multiple datasets, this work focuses on an automated workflow for improving message classification models used in an industrial operation for developing different ticketing systems. Although the results are about a single dataset in this paper, improvement in every class (e.g., the diagonal line in Figure 6(a)) was attainable, indicating the general applicability of the workflow. As the reviewer also indicated, data scarcity is a common issue, especially in the industry when new classes are introduced more commonly than academic datasets. Hence the workflow is generalizable as a highly useful (if not a necessary) process in such an industrial operation.
>
> References:
>
> TREC-6: https://cogcomp.seas.upenn.edu/Data/QA/QC/ (Li & Roth, 2022)
>
> Amazon Reviews’2023: https://amazon-reviews-2023.github.io/  (Hou et al., 2024)
>
> > **Q11 (W2): Additionally, the absences of baseline comparisons, for example, traditional data augmentation techniques, makes it challenging to fully understand the advantages of AutoGeTs.**
>
> **A11:** Thank you for reminding us about this comparison. We experimented with traditional data augmentation methods, the Easy Data Augmentation tool (EDA) [Wei & Zou, 2019] that implements synonym replacement, random insertion, random swap, random deletion, etc. We compared the performance of EDA and that of LLM through the experiments constrained by 1 GPU hour. One set of plots (objective: overall balanced accuracy or OBA) for the 15 classes is available in the “Revisions” folder of the updated Supplementary Materials, where four EDA lines (for SW, HSW, GA algorithms and random) are shown as solid lines, and four LLM lines are shown in dotted or dashed lines. We can observe that in all 15 plots, each solid line is below the corresponding dotted or dashed line (e.g., solid blue is below dotted blue). This indicated the merits of using LLMs. We will briefly describe the results at the beginning of Section 4.1 and detail the results in Appendix E.
>
> Reference:
>
> Wei, J. and Zou, K., 2019, November. EDA: Easy Data Augmentation Techniques for Boosting Performance on Text Classification Tasks. In *Proceedings of the 2019 Conference on Empirical Methods in Natural Language Processing and the 9th International Joint Conference on Natural Language Processing (EMNLP-IJCNLP)* (pp. 6382-6388).

---

> ### Author Response · Authors · 2024-11-20
> **This block addresses the last weakness.**
>
> > **Q12 (W3): While HSW and GA methods are intended to reduce computation costs, further analysis is needed to determine how computational demands scale with larger datasets, as real-world industrial applications typically involve substantial data volumes.**
>
> **A12:** Given n collected messages used for training, to select k messages from the n messages, there are C(n,k) = n!/[n!(n-k)!] combinations. To find the best example set is to search all combinations with different k, i.e., sum_{k=1}^n C(n,k). This is intractable computationally. Given a fixed time constraint, sliding window, HSW, and GA are finding different subsets of examples for prompting LLMs to generate different synthetic data. Our testing results showed that each of the three algorithms can perform better than the other two for classes with certain characteristics (as shown in Table 3). From an industrial perspective, one cannot rely on a simple answer, such as GA is the best or just use HSW. Hence, our research pointed to an alternative industrial approach, i.e., as described in Section 5.3, using an ensemble approach, placing the information obtained in the first iteration in a look-up table (a basic form of knowledge base), and selecting algorithms and metrics according to the look-up table for further iterations. For our application (text classification for ticketing systems), our contribution was the workflow for improving a classification model using synthetic data, which enables automatic formulation of the look-up table in the first iteration and use an ensemble algorithm for further iterations.
>
> This workflow can be applied to large datasets as well as small datasets. The datasets that we conducted experiments with are Inetum Research Data (39100 messages), TREC-6 (5452 questions) and Amazon Reviews’2023 (randomly selected 10000 reviews on Gift Cards). We will improve our description in Section 5.3 and our conclusions in Section 6 to make this contribution clearer.
>
> > **Q13: Reviewers PqjL(1) and qNuX(3) rated “fair” for the presentation and reviewer dzCV(2) found that numbers and fonts in experiment results figures are hard to read.**
>
> **A13:** Thank you. We have started to improve the presentation of the paper and increase the fonts in several figures.

---

> ### Author Response · Authors · 2024-11-25
> **Hope to Discuss Our Responses**
>
> Dear Reviewer qNuX,
>
> Thank you for your thorough review. We hope our responses have addressed your concerns, and we would greatly appreciate discussing them further as your insights are valuable. Please let us know if any clarifications are needed. Thank you again for your time and effort!
>
> Kind regards,
>
> Authors

---

### Official Review · Reviewer_dzCV · 2024-11-03

**Soundness:** 2
**Presentation:** 3
**Contribution:** 2
**Rating:** 5
**Confidence:** 3

**Summary:**

This paper studies the text classification task by exploring text data synthesis, which is especially useful to solve the data scarcity issue for long-tail classes. This paper specifically studies how to select demonstrations for generating synthetic data. Three approaches are proposed to select examples from a embedding space of existing data samples, including a naive traversal of embedding space with static sliding windows, hierarchical top-down search, and a generic algorithm to iteratively update the selection. Experiments results mainly shows that the data synthesis method can significantly improve model performance on the infrequent classes.

**Strengths:**

- This paper proposes three alternative methods for example selection from an embedding space to synthesize text data. Experiments compare these methods and their combinations to show their strength
- The proposed data synthesis method significantly improves classification performance on small-size classes, which shows the motivation of the paper
- Comprehensive experiments are conducted to compare the proposed three alternative methods and their combinations on different types of classes
- The paper overall is clearly written and easy to follow

**Weaknesses:**

- Only one dataset with skewed distributed classes are used. More datasets could be beneficial to show the generalizability of the method
- In the experiments, the proposed data synthesis methods are only compared with the baseline classifier. However, there should be a baseline with a "naive" example selection method such as random or using local density. I expect such a method is also effective to improve performance on rare classes and the performance increment of the proposed method may be marginal.
- The presentation of experiment results is hard to read. The numbers are too compact and hard to directly see the comparison. The figures have too many small fonts and cannot easily see the content. Some major revision here should be necessary.

**Questions:**

See the first two weaknesses above.

---

> ### Author Response · Authors · 2024-11-20
> **This block addresses the three weaknesses.**
>
> Thank you very much for summarizing and highlighting the strengths of this work. Below we focus on the issues raised. The reviewer’s queries are numbered by continuing that of Reviewer PqjL(1).
>
> > **Q7 (W1): Only one dataset with skewed distributed classes are used. More datasets could be beneficial to show the generalizability of the method**
>
> **A7:** We also experimented with other public datasets, such as TREC-6 (Li & Roth, 2022) and Amazon Reviews’2023 (Hou et al., 2024). We found that the improvement using synthetic data is in general feasible. However, the results as to the improvement of different classes were not comparable since the data and class labels are different. The quality of the baseline model is also a major factor. Unlike some research activities where a new model architecture, loss function, or pruning method was proposed and then tested with multiple datasets, this work focuses on an automated workflow for improving message classification models used in an industrial operation for developing different ticketing systems. Although the results are about a single dataset in this paper, improvement in every class (e.g., the diagonal line in Figure 6(a)) was attainable, indicating the general applicability of the workflow. As the reviewer also indicated, data scarcity is a common issue, especially in the industry when new classes are introduced more commonly than academic datasets. Hence the workflow is generalizable as a highly useful (if not a necessary) process in such an industrial operation.
>
> References:
>
> TREC-6: https://cogcomp.seas.upenn.edu/Data/QA/QC/ (Li & Roth, 2022)
>
> Amazon Reviews’2023: https://amazon-reviews-2023.github.io/  (Hou et al., 2024)
>
> > **Q8 (W2): In the experiments, the proposed data synthesis methods are only compared with the baseline classifier. However, there should be a baseline with a "naive" example selection method such as random or using local density. ...**
>
> **A8:** Thank you for reminding us about this comparison. In an early work (arXiv:2409.15848), where examples were manually selected. The comparison against random selection was carried out and briefly reported. We repeated the random selection testing with much more experiments involving all 15 classes. The optimization runs multiple iterations for 1 GPU hour, on average 299 iterations per class. A total of 4487 sets of randomly-selected examples were tested. We added the results as another benchmark. One set of plots (objective: overall balanced accuracy or OBA) for the 15 classes is available in the supplementary materials, where the random selection benchmark results are shown in gray dash-lines. We can observe that random-selection in general performed worse than the three algorithms (SW, HSW, GA) after 1 GPU hour. For classes T2 and T6, it performed slightly better than SW, and for T10, it performed worse than the original model. Below the 15 plots, we also provided the distributions of the amount of improvement or declination by the 4487 sets of random-selected examples, with overall performance (left) and class-based performance (right). The vertical black line indicates no change (neither improvement or declination). These figures are uploaded in “Q8-A8 LLM - Random Selection Baseline Results.jpg” in the “Revisions” folder of the Supplementary Materials. We will describe the random-testing results briefly at the beginning of Section 3.3 and detail the results in Appendix B.1.3.
>
> > **Q9 (W3): The presentation of experiment results is hard to read. …**
>
> **A9:** Thank you very much for pointing this out. We have started to increase the fonts in several figures. The new plots for Q8-A8 are examples of the improvement.

---

> ### Author Response · Authors · 2024-11-25
> **Hope to Discuss Our Responses**
>
> Dear Reviewer dzCV,
>
> Thank you for your thorough review. We hope our responses have addressed your concerns, and we would greatly appreciate discussing them further as your insights are valuable. We are actively improving the paper's presentation and will continue to do so. Please let us know if any clarifications are needed. Thank you again for your time and effort!
>
> Kind regards,
>
> Authors

---

### Official Review · Reviewer_PqjL · 2024-11-04

**Soundness:** 3
**Presentation:** 2
**Contribution:** 3
**Rating:** 5
**Confidence:** 4

**Summary:**

This paper explores the use of synthetic generations for retraining classification models with augmented data. The authors’ main contribution is through a selection pipeline for examples from the training set to be used in synthetic augmentation. They demonstrate their methodology through controlled experiments on class imbalanced settings.

**Strengths:**

- The paper demonstrates a good level of technical descriptions for experiments, definitions and results.
- Experiment setups are elaborate and controlled across different aspects of the selection methodology and throughout the retraining pipeline.
- Elaborate presentation of overall and class-specific performance results.
- The appendix is appreciated, especially the inclusion of baseline results and PCA experiments for synthetic correctness.

**Weaknesses:**

- A major issue for me is the lack of detail about synthetic generation in the main paper, this is an integral part of your contribution and yet it seems to be lacking in experimentation and presentation.
- There seems to be a major focus on the selection process but not as much exploration for the improvement of generation prompts/pipelines given these selections, in fact, according to the appendix, the synthetic generation prompt seems to be very vanilla, with little prompt engineering or any measures to maintain diversity or even any experimentation on the later. The lack of entropy injection or verification in the prompt could present an issue with generalizability and scale, these issues need to be addressed as part of synthetic pipelines not only with post-hoc checks.
- Section 5.2 could use further detail on how you obtain the order of classes optimized for R1 and R2 measures as well as relevant classes for ICs in R3, this is a difficult and important part of the problem and it would’ve been helpful if it was discussed further in the main paper.

**Questions:**

- Have you experimented with more models for generation?
- Have you performed any human annotations on a sample of your synthetic data in order to identify potential issues with diversity and correctness (is new data in fact in line with prompt examples)?

---

> ### Author Response · Authors · 2024-11-20
> **This block addresses the three weaknesses.**
>
> Thank you very much for providing such a concise and effective summary of the contributions of this work. Below we focus on the issues raised.
>
> > **Q1 (W1): A major issue for me is the lack of detail about synthetic generation in the main paper. …**
>
> **A1**: We will add new text in Section 3.1 to summarize the process of synthetic data generation (SDG), and provide further details in Appendix B.1, which currently contains the description of the parameters for SDG and an example.
>
> Our oversight was possibly due to that (i) we had been running the SDG process for so many times and forgot that this may not be obvious to readers, and (ii) we focused our attention on the process of example selection before SDG. SDG in a brief summary: (a) Evoke a new chat in GPT-3.5 through its API. Each message example is inserted into the prompt template as shown in Appendix B.1 (no context, zero-shot). The responses of the LLMs are cleaned, parsed, and separated automatically into <num=5> synthetic samples according to the formatting parameter in the prompt template.
>
>
> > **Q2 (W2): There seems to be a major focus on the selection process but not as much exploration for the improvement of generation prompts/pipelines given these selections, …**
>
> **A2**: As the reviewer correctly pointed out, the responses of LLMs are influenced by the selected examples as well as other LLM parameters in the prompt template. In addition, the parameters for training machine learning (ML) models also affect the observations about the effectiveness of LLM parameters. Because we focused our experiments on studying the three algorithms for example selection and the four metrics for guiding the algorithms, we had to ensure that other parameters are fixed by following the common guidelines in scientific experimentation. As reviewers know, the variation of other parameters would become the confounding effect that would undermine the observation and conclusions on the variations among the three algorithms and four metrics. While it is desired to conduct experiments with many variables (i.e., increasing entropy in experimentation), the total experimentation time required is <time per experiment> * K1 * K2 * K3 * … * Kn, where n is the number of variables and K[i] is the number of samples for the ith-variable. In this work, K1 = 3 (algorithms) and K2 = 4 (metrics). Because each experiment involves training and testing thousands of models (Figure 2), it is not feasible to vary other parameters at the same time. Nevertheless, we did conduct pilot studies to identify a suitable collection of LLM and ML parameters that are reasonably optimized for the three algorithms and four metrics. We believe that future research will provide observations and findings on other parameters.
>
> > **Q3 (W3): Section 5.2 could use further detail on how you obtain the order of classes optimized for R1 and R2 measures as well as relevant classes for ICs in R3, ...**
>
> **A3:** If we understand correctly, the reviewer considered the scenario that examples of class X are first used to optimize a model and then examples in class Y are used to optimize the model further. The data visualized in Figure 6 (Section 5.2) is about the first iteration (using examples of class X, X=T1, T2, T3, …). The goal is to observe if it can help improve class Y, T1, T2, T3, … and the overall model performance. This information was then used in the multi-iteration approach reported in Section 5.3. For example, if the objective is to improve the global performance of a model, the ensemble algorithm will choose classes in an order according to the performance data in Figure 6(b). If the objective is to improve the performance of Class X, the algorithm will choose classes (not necessarily Class X) in an order according to the data in Figure 6(a).
>
> Thank you for the suggestion. We will add additional texts in Sections 5.2 and 5.3 to improve the clarity, and will also add the visualization of the ordering of classes in a new Appendix C.2.

---

> ### Author Response · Authors · 2024-11-20
> **This block addresses the two questions.**
>
> > **Q4: Have you experimented with more models for generation?**
>
> **A4:** Yes, we have also experimented with using the Llama 3 instead of the GPT-3.5 as the LLM for generating synthetic samples in the early stage. We obtained similar results. We will add a few sentences in Appendix B.1 to describe the pilot study. We used GPT-3.5 in our large-scale experimentation as it was not feasible to add another variable about different language models. In addition, this work was built on the previous work within Inetum (Spain) for many years in developing ML models for ticketing systems. Inetum had experimented with a number of ML architectures and techniques. In general, the space of different models is gigantic (i.e., very high entropy). Fixing a model allows us to focus the experimentation on algorithms and metrics while reducing the confounding effects as much as possible. As the reviewer indicated, it is difficult to obtain meaningful observation based on sparse sampling of a high-entropy space. Note that although we focused on GPT-3.5, we had experimented with its parameter settings in our pilot study. Once these parameters were reasonably optimized, we fixed them in the large-scale experimentation for the reason described in Q2-A2.
>
> > **Q5: Have you performed any human annotations on a sample of your synthetic data in order to identify potential issues with diversity and correctness (is new data in fact in line with prompt examples)?**
>
> **A5:** In an early work (arXiv:2409.15848) where examples were manually selected, the examples were checked more frequently by authors who can read Spanish. In general, the number of the synthetic messages that were considered unusable was low, since many collected real-world messages contain language errors and convey confusing meanings.
>
> In this work, because the number of synthetic messages is excessively large, it is not feasible to check them. We relied on the effective setting used in the early manual work, and we checked the synthetic examples mostly for ensuring the scripts for running the experiments were correct (e.g., how many synthetic messages per example). Occasionally we randomly read some messages to check as part of such checks. We did not re-annotate any synthetic data as we had not encountered any synthetic message that belongs to a different class.
>
> > **Q6: Reviewers PqjL(1) and qNuX(3) rated “fair” for the presentation and reviewer dzCV(2) found that numbers and fonts in experiment results figures are hard to read.**
>
> **A6:** Thank you. We have started to improve the presentation of the paper and increase the fonts in several figures.

---

> ### Author Response · Authors · 2024-11-25
> **Hope to Discuss Our Responses**
>
> Dear Reviewer PqjL,
>
> Thank you for your thorough review. We hope our responses have addressed your concerns, and we would greatly appreciate discussing them further as your insights are valuable. Please let us know if any clarifications are needed. Thank you again for your time and effort!
>
> Kind regards,
>
> Authors

---

> > ### Comment · Reviewer_PqjL · 2024-11-26
> >
> > Thank you for your comprehensive and thoughtful responses to my concerns. I appreciate your time and effort in making modifications to address them, strengthening the portrayal of your research in the paper! Overall, I will maintain my rating.

---

### Author Response · Authors · 2024-11-28
**Interim Version Revisions Update**

Dear Reviewers,

We sincerely thank you for your time, effort, comments, and suggestions. While we continue improving the paper, we are submitting an interim version to report what has been done so far. The text revisions are marked in blue. Below, we outline the important revisions made.

1. In Section 3.1, we expanded the description of synthetic data generation (SDG) and provided further details in Appendix B.1 (including the SDG process, prompt template, format quality assurance of LLM responses, and parameter experiment results) to address Question 1 (W1) from Reviewer PqjL(1). Additionally in Appendix B.1, we mentioned the early experiments using Llama 3 for synthetic sample generation yielded comparable results as using GPT-3.5, addressing Question 4 (Q1) from the Reviewer PqjL(1).

2. In Section 3.3, we briefly described the limited performance of random example selection in the AutoGeTS workflow. We further detailed it in Appendix B.1.3, including its comparison with the three proposed example selection strategies (SW, HSW, and GA) under the same experimental setting. It can be considered as an additional baseline, addressing Q8 (W2) raised by Reviewer dzCV(2). The results show that random selection mostly underperforms the strategic methods, particularly for larger classes. Moreover, it yields only marginal overall improvements over the original model for classes T7, T13, T14, and T15, and even degrades the performance for class T10.

3. In Section 4.1, we briefly described the test of AutoGeTS workflow, in conjunction with the Easy Data Augmentation (EDA) tool [Wei & Zou,2019] for generating synthetic samples. EDA is a traditional data augmentation method. The testing results show that the EDA-based workflow underperforms in comparison with our LLM-based AutoGeTS workflow. We provided further comparative analysis in Appendix E, examining both the best improvements (overall and class-specific) and the temporal progression of overall balanced accuracy improvement across all 15 classes. The results show the LLM-based approach is always superior to the EDA-based approach, with this gap becoming larger when using strategic example selection methods (SW, HSW, GA) compared to random selection.

4. We have increased the line spacing in Table 2 in Section 4.2 and Table 3 in Section 5.1, and enlarged the font size in Figure 4 in Section 4.3 to enhance readability, addressing Question 9 (W3) raised by Reviewer dzCV(2). We have also checked the font size in all figures in appendices and improved those figures that had small fonts previously.

5. In Section 5.2, we clarified the class order determination for our ensemble algorithm based on Figure 6, and provided the detailed class orders for all three requirements (R1, R2, and R3) in Appendix C.2 with an illustrative Figure 28, addressing Question 3 (W3) from Reviewer PqjL(1).

6. In Section 5.3, we enhanced the presentation of the experimental results of ensemble AutoGeTS for improving class T13, and the discussion on the computational efficiency of HSW and GA methods. Accordingly, in Section 6 (Conclusions), we improved our presentation of the ensemble AutoGeTS algorithm and its contribution, addressing Question 12 (W3) from Reviewer qNuX(3).

---

### Author Response · Authors · 2024-12-04
**AutoGeTS Generalizability: New Results on TREC-6 Dataset**

Dear Reviewers,

We greatly appreciate your thorough review and valuable feedback. While we continue to enhance our paper based on your suggestions, we would like to share the results of our additional experiments using the TREC-6 dataset, which demonstrate the generalizability of our AutoGeTS workflow. These experiments specifically address the suggestions by Reviewer dzCV(2) (Q7-W1) and by Reviewer qNuX(3) (Q10-W1). The complete results can be found in a new appendix, Appendix F, at https://anonymous.4open.science/r/ICLR_2025_Submission_10435-8BB7/ICLR_AutoGeTS_Appendix%20F_TREC-6%20Results.pdf.

The TREC-6 dataset contains 5542 fact-based questions distributed across six semantic classes, with varying class sizes from 86 to 1250 questions. Likely, because the overall size of the dataset is much smaller than the ticketing dataset used in our main experimentation, both EDA-based and LLM-based workflows achieved improvements compared to the baseline M0 model across different testing conditions, i.e.,

- selection strategies (random, SW, HSW, GA),
- objectives (6 classes, overall)
- metrics (global OBA, local CBA)

Meanwhile, our three selection methods (SW, HSW, and GA) showed better performance than random selection, with the LLM-based workflow performs consistently better than the EDA-based approach across all testing conditions.

---

### Meta-Review · Area_Chair_UHyD · 2024-12-18

**Metareview:**

The authors introduce Automated Generation of Text Synthetics (AutoGeTS), an approach to synthetic data generation for text classification tasks.

While this seems to represent a reasonable approach, the technical contribution is rather limited. This could be sufficient if the empirical evaluation were more thorough, but as is the evaluation relies on a single dataset (though a second was added in rebuttal, this is still only two datasets, which seems inadequate to me), and there is a lack of appropriate augmentation baselines (a point partially addressed in rebuttal, although only EDA was considered). These aspects greatly limit the scope of the contribution.

**Additional Comments On Reviewer Discussion:**

The authors did address some of the main questions in discussion. For example, in response to dzCV's point that only a single dataset was used, the authors say they did experiment with two other classification datasets but and claim "We found that the improvement using synthetic data is in general feasible"; it is not really clear what this means, at least to me. (The authors add a link to TREC-6 results, but this does not address the fundamental concern about generalizability based on now two datasets.)

Other points of clarification from the authors were welcome.

---

### Decision · Program_Chairs · 2025-01-22

Reject